# Temporal Causal Mediation through a Point Process: Direct and Indirect Effects of Healthcare Interventions

**Çağlar Hızlı**[*]
Dept. of Computer Science
Aalto University

**ST John**
Dept. of Computer Science
Aalto University

**Anne Juuti**
Helsinki University Hospital
and University of Helsinki

**Tuure Saarinen**
Helsinki University Hospital
and University of Helsinki

**Kirsi Pietiläinen**
Helsinki University Hospital
and University of Helsinki

**Pekka Marttinen**
Dept. of Computer Science
Aalto University

## Abstract

Deciding on an appropriate intervention requires a causal model of a treatment, the outcome, and potential mediators. Causal mediation analysis lets us distinguish between direct and indirect effects of the intervention, but has mostly been studied in a static setting. In healthcare, data come in the form of complex, irregularly sampled time-series, with dynamic interdependencies between a treatment, outcomes, and mediators across time. Existing approaches to dynamic causal mediation analysis are limited to regular measurement intervals, simple parametric models, and disregard long-range mediator–outcome interactions. To address these limitations, we propose a non-parametric mediator–outcome model where the mediator is assumed to be a temporal point process that interacts with the outcome process. With this model, we estimate the direct and indirect effects of an external intervention on the outcome, showing how each of these affects the whole future trajectory. We demonstrate on semi-synthetic data that our method can accurately estimate direct and indirect effects. On real-world healthcare data, our model infers clinically meaningful direct and indirect effect trajectories for blood glucose after a surgery.

## 1  Introduction

In healthcare, a key challenge is to design interventions that effectively control a target outcome [Vamathevan et al., 2019, Ghassemi et al., 2020]. To design an efficient intervention, decision-makers need not only to estimate the *total* effect of the intervention, but also understand the underlying causal mechanisms driving this effect. To this end, causal mediation analysis decomposes the total effect into: *(i)* an *indirect* effect flowing through an intermediate variable (*mediator*), and *(ii)* a *direct* effect representing the rest of the effect. The statistical question is then to estimate the direct and indirect effects.

As a running example, we consider the effect of bariatric surgery (*treatment*) on meal–blood glucose (*mediator–outcome*) dynamics. In Fig. 1, we show how blood glucose changes after the surgery. This change is partly mediated by the changes in diet before and after the surgery (alterations in meal size and frequency). However, the surgery can also directly impact blood glucose levels (via metabolic processes, regardless of meal size). To efficiently control glucose, we need to estimate to what extent the effect of surgery is due to the post-surgery diet (*indirect*) or other metabolic processes (*direct*).

Causal mediation has been studied extensively in a static setting with three variables: a treatment $A$, a mediator $M$, and an outcome $Y$ [Robins and Greenland, 1992, Pearl, 2001, Robins, 2003, Vander-Weele, 2009, Imai et al., 2010a,b, Tchetgen and Shpitser, 2012]. However, as the running example

---

[*]Correspondence: `caglar.hizli@aalto.fi`

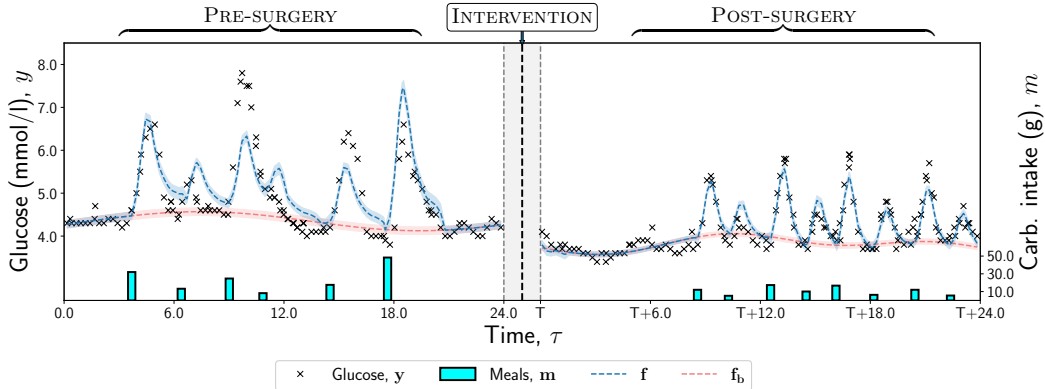

Figure 1: Comparison of the meal–blood glucose dynamics of one patient for the pre- and post-surgery periods: meals (carbohydrate intake, cyan bars), glucose (black crosses), predicted glucose baseline $\mathbf{f_b}$ (red dashed line) and predicted glucose progression $\mathbf{f}$ (blue dashed line). For blood glucose, we see, *after the surgery*, (i) the baseline $\mathbf{f_b}$ declines (red dashed line, pre-surgery $\rightarrow$ post-surgery), and (ii) the glucose response of a meal has a steeper rise and a steeper decline (blue peaks, pre-surgery $\rightarrow$ post-surgery). For meals, we see, *after the surgery*, the patient eats (i) more frequently and (ii) less carbohydrates (cyan bars, pre-surgery $\rightarrow$ post-surgery). A relevant research question is: how much of the surgery effect is due to the post-surgery diet (*indirect*) or other biological processes (*direct*)?

illustrates, many real-world healthcare problems are dynamic, i.e., the variables evolve over time and are measured at multiple time points, e.g., a patient's state observed through electronic health records (EHR) [Saeed et al., 2011, Soleimani et al., 2017, Schulam and Saria, 2017], tumour growth under chemotherapy and radiotherapy [Geng et al., 2017, Bica et al., 2020, Seedat et al., 2022], and continuous monitoring of blood glucose under meal and insulin events [Berry et al., 2020, Wyatt et al., 2021, Hızlı et al., 2023]. Hence, instead of observing a single mediator and a single outcome after an intervention, in many situations the intervention is followed by *(i) sequences* of mediators and outcomes that vary over time. To complicate the matters further, these sequences may be complex, e.g., *(ii)* sparse and irregularly-sampled, as in EHR, or *(iii)* with long-range mediator–outcome dependencies.

Some existing works have addressed dynamic causal mediation for *(i)* a sequence of mediators and outcomes [VanderWeele and Tchetgen Tchetgen, 2017, Lin et al., 2017, Zheng and van der Laan, 2017, Vansteelandt et al., 2019, Aalen et al., 2020]; however, they are limited to measurements at regular time intervals. Furthermore, they are based on the parametric g-formula [Robins, 1986] and marginal structural models [Robins et al., 2000a], which can handle time-varying confounding but rely on simple linear models. A recent work by Zeng et al. [2021] (Fig. 2a) studied causal mediation for *(ii)* sparse and irregular longitudinal data, but considered neither *(iii)* long-range mediator–outcome dependencies nor the interaction between past outcomes and future mediators ($\cdots\triangleright$, Fig. 2c).

We address these limitations in dynamic causal mediation by introducing a method where both the mediator as well as the outcome are stochastic processes (as opposed to single variables in the static case). In this setup, we provide the estimated direct and indirect effects as longitudinal counterfactual trajectories, and theoretically, we present causal assumptions required to identify them. We model the mediator sequence as a marked point process (MPP), and the outcome sequence as a continuous-valued stochastic process, building on a non-parametric model [Hızlı et al., 2023]. This allows for irregularly-sampled measurements of mediator and outcome processes, while capturing non-linear, long-range interactions between them. Recently, a different line of research investigated causal inference for MPPs [Gao et al., 2021, Zhang et al., 2022, Noorbakhsh and Rodriguez, 2022], but they have not considered a causal mediation setting. Compared to the existing literature [Hızlı et al., 2023], our model enables the estimation of the direct and indirect effects separately by incorporating an external intervention that jointly affects both mediator and outcome processes (Fig. 2b vs. Fig. 2c).

Our contributions are as follows:

- **Dynamic causal mediation with a point process mediator.** We generalize the dynamic causal mediation problem by modeling the mediator as a point process, which can have a ran-

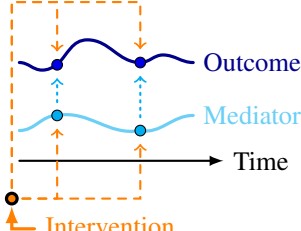

(a) Direct and indirect effects with non-interacting longitudinal mediators [Zeng et al., 2021].

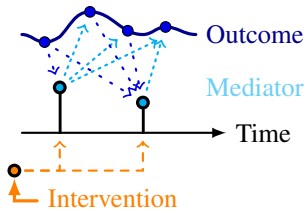

(b) Fully-mediated policy effect with interacting point process mediators [Hızlı et al., 2023].

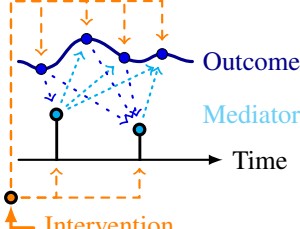

(c) Direct and indirect effects with interacting point process mediators (Our work).

Figure 2: Comparison of available implementations for dynamic causal mediation. **(a)** Zeng et al. [2021] studied dynamic causal mediation for a sparse and irregularly-sampled mediator–outcome sequence; their counterfactual formulation allows for only uni-directional effects from mediator to outcome (·····>), while their implementation assumes the effects are instantaneous. **(b)** Hızlı et al. [2023] considered the long-range interaction between mediator and outcome processes (·····> + ···>), but did not allow the intervention to directly affect the outcome. **(c)** While capturing long-range dependencies, our work allows for an intervention that affects both processes jointly (- - >).

dom number of occurrences at random times and complex interactions with the outcome. Furthermore, we provide causal assumptions required to identify the direct and indirect effects.

- **A mediator–outcome model with an external intervention.** We extend a previous non-parametric model for mediators and outcomes [Hızlı et al., 2023] to include an external intervention that affects the joint dynamics of these two processes.

- In a real-world experiment, we study in a data-driven manner **the effect of bariatric surgery for treatment of obesity on blood glucose dynamics**, and separately estimate the direct effect from the indirect effect whereby surgery affects blood glucose through changed diet.

## 2 Dynamic Causal Mediation

In this section, we formulate the dynamic causal mediation problem for a treatment process $A$, a mediator process $M$ and an outcome process $Y$, and contrast this with the static causal mediation definition for single variables $A$, $M$ and $Y$. Our formulation builds on structural causal models [SCM, Pearl, 2009] and an interventionist approach to causal mediation [Robins et al., 2022].

**Structural Causal Model.** An SCM $\mathcal{M} = (\mathbf{S}, p(\mathbf{U}))$ of a set of variables $\mathbf{X} = \{X_k\}_{k=1}^K$ is defined as a pair of (i) *structural equations* $\mathbf{S} = \{X_k := f_k(\mathrm{pa}(X_k), U_k)\}_{k=1}^K$ and (ii) a *noise distribution* $\mathbf{U} = \{U_k\}_{k=1}^K \sim p(\mathbf{U})$ [Pearl, 2009, Peters et al., 2017]. The SCM entails an observational distribution $\mathbf{X} \sim p(\mathbf{X})$. To see this, consider sampling noise $\mathbf{U} \sim p(\mathbf{U})$ and then performing a forward pass through the structural equations $\mathbf{S}$. In the SCM, an intervention $\mathrm{do}(X_k = \tilde{x}_k)$ is defined as an operation that removes the structural equation $f_k(\cdot, \cdot)$ corresponding to $X_k$ and sets $X_k$ to the value $\tilde{x}_k$, yielding an interventional distribution $p(\mathbf{X} \mid \mathrm{do}(X_k = \tilde{x}_k))$.

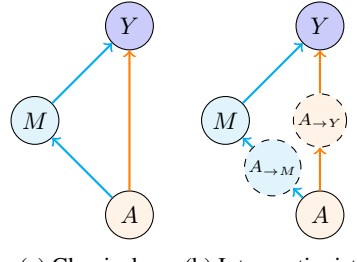

(a) Classical.    (b) Interventionist.

Figure 3: Graphical models for causal mediation. Mediation is defined in **(a)** using nested counterfactuals, while in **(b)** using path-specific treatments $A_{\to M}$ and $A_{\to Y}$.

**Interventionist Approach to Causal Mediation.** Consider an SCM of a treatment $A$, a mediator $M$ and an outcome $Y$ (Fig. 3a). For brevity, we focus on the indirect path, but the same reasoning applies for the direct path. To estimate the indirect effect, we want to intervene *only* on the indirect path ($A{\to}M{\to}Y$), while keeping the direct path ($A{\to}Y$) at its natural value. To achieve this, we follow an interventionist approach [Robins et al., 2022]. We hypothetically duplicate the treatment $A$ into an indirect-path treatment $A_{\to M}$ and a direct-path treatment $A_{\to Y}$, so that each of these affects *only* its target path (Fig. 3b). For instance, in the surgery–diet–glucose example (*treatment–mediator–outcome*), the indirect-path treatment

$A_{\to M}$ corresponds to changing *only* the diet, without changing how the surgery affects blood glucose through other metabolic processes. Using path-specific interventions, we follow Pearl [2001] and define the natural indirect effect (NIE) as

$$\text{NIE}(\tilde{a}) \equiv Y[A_{\to Y} = \varnothing, A_{\to M} = \tilde{a}] - Y[A_{\to Y} = \varnothing, A_{\to M} = \varnothing] \equiv Y[\varnothing, \tilde{a}] - Y[\varnothing, \varnothing], \quad (1)$$

where we use the short-hand notation for an intervention $[\tilde{a}_1, \tilde{a}_2] \equiv \text{do}(A_{\to Y} = \tilde{a}_1, A_{\to M} = \tilde{a}_2)$, and $Y[\tilde{a}_1, \tilde{a}_2]$ is the *potential* outcome after setting the direct-path treatment (first argument) to $\tilde{a}_1$ and the indirect-path treatment to $\tilde{a}_2$. The value $\varnothing$ means that the corresponding pathwise treatment is set to its *natural* value, i.e., the value the variable would have attained without our doing any intervention. Hence, the indirect effect is intuitively defined in Eq. (1) as the difference of the outcome $Y$ when the direct-path treatment is fixed to $\varnothing$ while the indirect-path treatment is either $\tilde{a}$ or $\varnothing$.

The natural direct effect (NDE) is similarly defined as [Pearl, 2001, Robins et al., 2022]

$$\text{NDE}(\tilde{a}) \equiv Y[A_{\to Y} = \tilde{a}, A_{\to M} = \tilde{a}] - Y[A_{\to Y} = \varnothing, A_{\to M} = \tilde{a}] \equiv Y[\tilde{a}, \tilde{a}] - Y[\varnothing, \tilde{a}], \quad (2)$$

However, these definitions are limited to the static case with single variables $A$, $M$ and $Y$. Hence, they are not applicable as such to complex, real-world dynamics of healthcare data sets, where a treatment is typically followed by sequences of irregularly-sampled mediator and outcome measurements.

## 2.1 Direct and Indirect Effects as Stochastic Processes

We extend the static definitions of natural direct and indirect effects by formulating them as stochastic processes. In the following, we assume a single patient is observed over a period $[0, T]$. The formulation trivially generalizes to multiple patients, assuming exchangeability.

For each patient, we define a mediator process $\boldsymbol{M}$ and an outcome process $\boldsymbol{Y}$. We assume the treatment $A$ represents a single intervention occurring once at time $t_a \in [0, T]$, e.g., a surgery. For completeness, we define also the treatment as a stochastic process $\boldsymbol{A}$ corresponding to a binary counting process $\boldsymbol{N}_A : [0, T] \to \{0, 1\}$ that specifies the pre- and post-intervention periods. The treatment is applied to control an outcome of interest over time, e.g. continuously-monitored blood glucose, which is defined as the outcome process $\boldsymbol{Y} : [0, T] \to \mathbb{R}$. We assume the treatment affects the outcome process $\boldsymbol{Y}$ via two separate causal paths, where the indirect path ($\cdots\!\!\succ$, Fig. 2c) that goes through mediators occurring at non-deterministic times, e.g., meals, is of special interest. Hence, we define this causal path explicitly through the mediator process $\boldsymbol{M} : [0, T] \to \mathbb{N} \times \{\mathbb{R} \cup \{\varnothing\}\}$, such that $M(\tau) = (k, m)$, where $k$ is the number of occurrences of the mediating event up until time $\tau$ and $m$ is the value of the mediator at time $\tau$. Occurrences $k$ follow a counting process $\boldsymbol{N}_M : [0, T] \to \mathbb{N}$ and values $m$ follow a dosage process $\boldsymbol{D}_M : [0, T] \to \{\mathbb{R} \cup \{\varnothing\}\}$, such that these two processes contain all the information about the mediator process on the period $[0, T]$. At occurrence times, the mediator gets a value $m \in \mathbb{R}$ and at other times $m = \varnothing$. The rest of the possible causal paths are not modeled explicitly but instead their effect is jointly represented as the direct path $A \to \boldsymbol{Y}$ ($-\!-\!\succ$, Fig. 2c).

We define direct and indirect treatment effects formally analogously to the static case, and assume that the treatment process $\boldsymbol{A}(\tau)$ is hypothetically duplicated into the direct-path treatment $\boldsymbol{A}_{\to Y}(\tau)$ and the indirect-path treatment $\boldsymbol{A}_{\to M}(\tau)$ [Robins and Richardson, 2010, Didelez, 2019]. The indirect-path treatment $\boldsymbol{A}_{\to M}(\tau)$ affects the outcome only through the mediator process $\boldsymbol{M}$. Similarly, the direct-path treatment $\boldsymbol{A}_{\to Y}(\tau)$ affects the outcome through processes other than the mediator.

In practice, we assume that we observe the mediator process $\boldsymbol{M}$ at $I$ time points: $\{(t_i, m_i)\}_{i=1}^{I}$, where $t_i$ is the occurrence time of the $i^{\text{th}}$ mediator event and $m_i$ the corresponding value. Similarly, we observe the outcome process $\boldsymbol{Y}$ as $\{(t_j, y_j)\}_{j=1}^{J}$. We observe $\boldsymbol{A}(\tau)$ as the occurrence of a single treatment event at time $t_a \in [0, T]$. In the 'factual' world, treatments $A$, $A_{\to Y}$ and $A_{\to M}$ occur at the same time, i.e., their respective counting processes coincide:

$$\boldsymbol{N}_A(\tau) = \boldsymbol{N}_{A_{\to Y}}(\tau) = \boldsymbol{N}_{A_{\to M}}(\tau) = \mathbb{1}\{\tau \geq t_a\}. \quad (3)$$

For instance, in the surgery–diet–blood glucose (*treatment–mediator–outcome*) example, the surgery starts to affect both the diet and the other metabolic processes immediately after it has occurred at time $t_a$. To measure how the surgery affects the blood glucose (indirectly) through diet or (directly) through other metabolic processes, we consider a hypothetical intervention that changes *either* the diet *or* the other metabolic processes, but not both at the same time. Formally, we devise these hypothetical interventions that activate *only* the direct or the indirect causal path by setting treatment times $t_{A_{\to Y}}$ and $t_{A_{\to M}}$ to distinct values.

Using these hypothetical interventions, we formulate two causal queries *to understand how the treatment affects the outcome process via direct and indirect casual mechanisms*. The natural indirect effect (NIE) is defined as

$$\text{NIE}(\tilde{t}_a) \equiv \boldsymbol{Y}_{>\tilde{t}_a}[A_{\to Y} = \varnothing, A_{\to M} = \tilde{t}_a] - \boldsymbol{Y}_{>\tilde{t}_a}[A_{\to Y} = \varnothing, A_{\to M} = \varnothing]$$
$$\equiv \boldsymbol{Y}_{>\tilde{t}_a}[\varnothing, \tilde{t}_a] - \boldsymbol{Y}_{>\tilde{t}_a}[\varnothing, \varnothing]. \tag{4}$$

A close comparison with the static case in Eq. (1) reveals that the outcome is here defined as a stochastic process $\boldsymbol{Y}_{>\tilde{t}_a}$ which describes its entire future trajectory; i.e., a sample from this process corresponds to an entire sequence of values representing the continuation of the outcome after the intervention at time $\tilde{t}_a$. For example, Eq. (4) can quantify the contribution of the change of diet at time $\tilde{t}_a$ on the blood glucose progression, while other metabolic processes are fixed at their natural state, $\varnothing$, before the surgery. Similarly, we estimate the natural direct effect (NDE) using

$$\text{NDE}(\tilde{t}_a) \equiv \boldsymbol{Y}_{>\tilde{t}_a}[A_{\to Y} = \tilde{t}_a, A_{\to M} = \tilde{t}_a] - \boldsymbol{Y}_{>\tilde{t}_a}[A_{\to Y} = \varnothing, A_{\to M} = \tilde{t}_a]$$
$$\equiv \boldsymbol{Y}_{>\tilde{t}_a}[\tilde{t}_a, \tilde{t}_a] - \boldsymbol{Y}_{>\tilde{t}_a}[\varnothing, \tilde{t}_a], \tag{5}$$

which quantifies the contribution of the changes in other metabolic processes on the blood glucose progression, while the diet is fixed at its natural state after the surgery. The total effect of the intervention $[t_a = \tilde{t}_a]$ is equal to the sum of the NDE and the NIE:

$$\text{TE}(\tilde{t}_a) = \text{NDE}(\tilde{t}_a) + \text{NIE}(\tilde{t}_a) = \boldsymbol{Y}_{>\tilde{t}_a}[\tilde{t}_a, \tilde{t}_a] - \boldsymbol{Y}_{>\tilde{t}_a}[\varnothing, \varnothing]. \tag{6}$$

## 3  Causal Assumptions and Identifiability

In this section, we first develop a mathematical formulation of the causal assumptions required to identify direct and indirect effect trajectories $\text{NDE}(\tilde{t}_a)$ and $\text{NIE}(\tilde{t}_a)$. Under these assumptions, we next represent the causal queries in the form of statistical terms, which we can estimate using a mediator–outcome model. The proofs and further details are presented in Appendix A.

To identify the direct, indirect and total effects, we make the following Assumptions (**A1**, **A2**, **A3**):

$$\textbf{(A1):} \quad p(\cdot \mid \text{do}(\boldsymbol{A} = \tilde{\boldsymbol{N}}_A)) \equiv p(\cdot \mid \boldsymbol{A} = \tilde{\boldsymbol{N}}_A),$$
$$\textbf{(A2):} \quad \boldsymbol{A}_{\to Y} \perp\!\!\!\perp M(\tau) \mid \boldsymbol{A}_{\to M}, \mathcal{H}_{<\tau}, \quad \tau \in [0, T],$$
$$\textbf{(A3):} \quad \boldsymbol{A}_{\to M} \perp\!\!\!\perp Y(\tau) \mid \boldsymbol{A}_{\to Y}, \mathcal{H}_{<\tau}, \quad \tau \in [0, T],$$

where $\mathcal{H}_{<\tau}$ is the history up to time $\tau$, including both mediators and outcomes. **(A1)** is a continuous-time version of the No-Unobserved-Confounders assumption for process pairs $(\boldsymbol{A}, \boldsymbol{M})$ and $(\boldsymbol{A}, \boldsymbol{Y})$ [continuous-time NUC, Schulam and Saria, 2017]. **(A2**, **A3)** ensure that path-specific treatments $A_{\to M}$ and $A_{\to Y}$ causally affect only their target path, conditioned on the past. Accordingly, they imply no unobserved confounding between mediator and outcome processes $(\boldsymbol{M}, \boldsymbol{Y})$. **(A1**, **A2**, **A3)** might not hold in observational studies and they are not statistically testable. We discuss their applicability in the running example in Section 5.1.

For the total effect, we need to estimate the trajectories $\boldsymbol{Y}_{>\tilde{t}_a}[\tilde{t}_a, \tilde{t}_a]$ and $\boldsymbol{Y}_{>\tilde{t}_a}[\varnothing, \varnothing]$, which are identifiable under **(A1)**. For direct and indirect effects, we further need to estimate the counterfactual trajectory $\boldsymbol{Y}_{>\tilde{t}_a}[\varnothing, \tilde{t}_a]$, i.e., the outcome process under a hypothetical intervention, which is identifiable under **(A1**, **A2**, **A3)**. We consider a counterfactual trajectory $\boldsymbol{Y}_{>\tilde{t}_a}$ under a paired intervention, e.g. $[\varnothing, \tilde{t}_a]$, at $R$ ordered query points $\mathbf{q} = \{q_r\}_{r=1}^R$, $q_r > \tilde{t}_a$: $\boldsymbol{Y}_{\mathbf{q}}[\varnothing, \tilde{t}_a] = \{Y(q_r)[\varnothing, \tilde{t}_a]\}_{r=1}^R$. Provided that **(A1**, **A2**, **A3)** hold, the interventional distribution of the counterfactual trajectory $\boldsymbol{Y}_{\mathbf{q}}[\varnothing, \tilde{t}_a]$ is given by

$$P(\boldsymbol{Y}_{\mathbf{q}}[\varnothing, \tilde{t}_a]) = \sum_{\boldsymbol{M}_{>\tilde{t}_a}} \prod_{r=0}^{R-1} \underbrace{P(Y_{q_{r+1}} | M_{[q_r, q_{r+1}]}, t_a = \varnothing, \mathcal{H}_{\leq q_r})}_{\text{outcome term } \bullet} \underbrace{P(M_{[q_r, q_{r+1}]} | t_a = \tilde{t}_a, \mathcal{H}_{\leq q_r})}_{\text{mediator term } \circ}, \tag{7}$$

where $\boldsymbol{M}_{>\tilde{t}_a} = \cup_{r=0}^{R-1} M_{[q_r, q_{r+1}]}$. The two terms in Eq. (7) can be estimated by an interacting mediator–outcome model, discussed in Section 4. In addition, the definitions of the $\text{NIE}(\tilde{t}_a)$ and the $\text{NDE}(\tilde{t}_a)$ in Eqs. (4) and (5) under a single treatment at time $\tilde{t}_a$ can be extended to direct and indirect effects under a sequence of healthcare interventions, if necessary. We further illustrate this extension in Appendix A.3 and provide an identifiability analysis in such multiple treatment setups.

# 4 Interacting Mediator–Outcome Model

To model the interacting mediator–outcome processes, we extend a non-parametric model for mediators and outcomes [Hızlı et al., 2023] to include an external intervention $A$ that affects the joint dynamics of these two processes. Similary, we combine a marked point process [MPP, Daley and Vere-Jones, 2003] and a conditional Gaussian process [GP, Williams and Rasmussen, 2006]. For background on MPP, GP, and details on model definitions, inference and scalability, see Appendix B.

Each patient is observed in two regimes: $\mathcal{D} = \{\mathcal{D}^{(a)}\}_{a \in \{0,1\}}$. Within each regime $a \in \{0,1\}$, the data set $\mathcal{D}^{(a)}$ contains the measurements of mediator $M$ and outcome $Y$ at irregular times: $\{(t_i^{(a)}, m_i^{(a)})\}_{i=1}^{I^{(a)}}$ and $\{(t_j^{(a)}, y_j^{(a)})\}_{j=1}^{J^{(a)}}$. For the full data set $\mathcal{D}$, the joint distribution is

$$p(\mathcal{D}) = \prod_{a \in \{0,1\}} \left[ \exp(-\Lambda^{(a)}) \prod_{i=1}^{I^{(a)}} \underbrace{\lambda^{(a)}(t_i, m_i \mid \mathcal{H}_{<t_i})}_{\text{mediator intensity } \circ} \prod_{j=1}^{J^{(a)}} \underbrace{p^{(a)}(y_j \mid t_j, \mathcal{H}_{<t_j})}_{\text{outcome model } \bullet} \right], \quad (8)$$

where we denote the conditional intensity function of the mediator MPP by $\lambda^{(a)}(t_i, m_i \mid \mathcal{H}_{<t_i})$, and the point process integral term by $\Lambda^{(a)} = \int_0^T \lambda^{(a)}(\tau \mid \mathcal{H}_{<\tau}) \, d\tau$.

## 4.1 Mediator Intensity (○)

The mediator intensity $\lambda(t_i, m_i \mid \mathcal{H}_{<t_i})$ is defined as a combination of the mediator time intensity $\lambda(t_i \mid \mathcal{H}_{<t_i})$ and the mediator mark or dosage density $p(m_i \mid t_i, \mathcal{H}_{<t_i})$: $\lambda(t_i, m_i \mid \mathcal{H}_{<t_i}) = \lambda(t_i \mid \mathcal{H}_{<t_i}) p(m_i \mid t_i, \mathcal{H}_{<t_i})$. Similar to [Hızlı et al., 2023], we model the time intensity $\lambda(t_i \mid \mathcal{H}_{<t_i})$ as the squared sum of three components $\{\beta_0, g_m, g_o\}$, to ensure non-negativity:

$$\lambda(\tau \mid \mathcal{H}_{<\tau}) = \Big( \underbrace{\beta_0}_{\substack{\text{PP} \\ \text{baseline}}} + \underbrace{g_m(\tau; \mathbf{m})}_{\substack{\text{mediator} \\ \text{effect}}} + \underbrace{g_o(\tau; \mathbf{o})}_{\substack{\text{outcome} \\ \text{effect}}} \Big)^2, \quad (9)$$

where $\mathbf{m}$ contains past mediators (meal times and sizes) and $\mathbf{o}$ contains past outcomes (glucose). The constant $\beta_0$ serves as a basic Poisson process baseline. The mediator-effect function $g_m$ and the outcome-effect function $g_o$ model how the intensity depends on the past mediators and outcomes, respectively. The components $g_m$ and $g_o$ are time-dependent functions with GP priors: $g_m, g_o \sim \mathcal{GP}$. The mediator dosage density $p(m_i \mid t_i, \mathcal{H}_{<t_i})$ is also modeled with an independent GP prior: $m_i(t_i) \sim \mathcal{GP}$.

## 4.2 Outcome Model (●)

We model the outcome process $Y = \{Y(\tau \mid \mathcal{H}_{<\tau}) : \tau \in [0, T]\}$ as a conditional GP prior, that is, a sum of three independent functions [Schulam and Saria, 2017, Hızlı et al., 2023]:

$$Y(\tau \mid \mathcal{H}_{<\tau}) = \underbrace{f_b(\tau)}_{\text{baseline}} + \underbrace{f_m(\tau; \mathbf{m})}_{\text{mediator response}} + \underbrace{u_Y(\tau)}_{\text{noise}}, \quad (10)$$

where the baseline and the mediator response have GP priors, $f_b, f_m \sim \mathcal{GP}$, similar to Hızlı et al. [2023], while the noise is sampled independently from a zero-mean Gaussian variable $u_Y(\tau) \sim \mathcal{N}(0, \sigma_y^2)$. The mediator response function $f_m$ models the dependence of the future outcomes on the past mediators. We assume that the effects of nearby mediators simply sum up: $f_m(\tau; \mathbf{m}) = \sum_{i:t_i \leq \tau} l(m_i) f_m^0(\tau - t_i)$, where $f_m^0 \sim \mathcal{GP}$ describes a shared shape function and the magnitude of the effect depends on the mark $m_i$ through a function $l(m_i)$.

# 5 Experiments

In this section, we validate the ability of our method to separate direct and indirect effects of a healthcare intervention on interacting mediator–outcome processes. First, we show that our model can support clinical decision-making in *real-world data* from a randomized controlled trial (RCT) about the effect of gastric bypass surgery (*treatment*) on meal–blood glucose (*mediator–outcome*) dynamics. Second, as the true causal effects are unknown in the real-world data, we set up a *realistic, semi-synthetic simulation study*, where we evaluate the performance of our model on two causal tasks: estimating direct and indirect effect trajectories. Our implementation and code to reproduce the study can be found at `https://github.com/caglar-hizli/dynamic-causal-mediation`.

## 5.1 Real-World Study

We first show on data from a real-world Rᴄᴛ [Saarinen et al., 2019, Ashrafi et al., 2021] that our model can learn clinically-meaningful direct and indirect effects of bariatric surgery on blood glucose, by analyzing how surgery affects glucose through changed diet (indirectly) or other metabolic processes (directly). For further details, see Appendix C.1.

**Dataset.** For 15 nondiabetic obesity patients (body mass index, Bᴍɪ $\geq 35\,\mathrm{kg/m^2}$) undergoing a gastric bypass surgery, a continuous-monitoring device measured the blood glucose of each patient at 15-minute intervals. Patients recorded their meals in a food diary, and the nutrient information was processed into total carbohydrates (sugar + starch) for each meal. Data was collected over two 3-day long periods: (i) *pre-surgery* and (ii) *post-surgery*.

**Causal Assumptions.** Assumption (**A1**) implies a treatment that is non-informative. The surgery does not depend on patient characteristics (each patient undergoes the surgery), therefore (**A1**) holds. If this was not the case, we could have used a debiasing method such as inverse probability treatment weighting [Robins et al., 2000a]. Assumptions (**A2**, **A3**) imply that (i) path-specific treatments $A_{\to M}$ and $A_{\to Y}$ affect only their target path and (ii) there is no unobserved confounding between mediators and outcomes. They do **not** exactly hold in the study. For (i), one way to realize $A_{\to M}$ is a dietary intervention that only changes the diet; however, it might not be possible to change the rest of the metabolic processes through $A_{\to Y}$ without affecting the diet. For (ii), demographic factors such as age, gender, body weight, and height could be potential confounder variables that could bias the estimation of surgery effects. Moreover, there can be unobserved processes such as exercising habits that affect both meals and blood glucose, even though the impact of confounders other than meals on blood glucose is likely modest for nondiabetics, since blood glucose is known to be relatively stable between meals [Ashrafi et al., 2021].

### 5.1.1 Bariatric Surgery Regulates Glucose Through Meals and Other Metabolic Processes

To visualize the effect of the surgery on meal–blood glucose dynamics, we show 1-day long meal–glucose measurements for one patient together with predicted glucose trajectories from our outcome model in both pre- and post-surgery periods in Fig. 1. For blood glucose, we see, *after the surgery*, (i) the baseline declines, and (ii) a meal produces a faster rise and decline. For meals, we see, *after the surgery*, the patient eats (i) more frequently and (ii) less carbohydrate per meal.

### 5.1.2 Direct and Indirect Effects of the Surgery on Meal Response Curves

We compare the predicted glucose baseline $f_b$ and the predicted meal response $f_m$ for all patients in Fig. 4. In Fig. 4a, *after the surgery*, the glucose baseline declines in agreement with previous findings [Dirksen et al., 2012, Jacobsen et al., 2013]. In Fig. 4b, we see meal responses corresponding to four hypothetical scenarios. In the **upper-left**, we predict a typical *pre-surgery* meal response and highlight the area under this curve by shaded grey. In the **upper-right**, we predict the hypothetical meal response by considering *only* the surgical effect on metabolism, while keep-

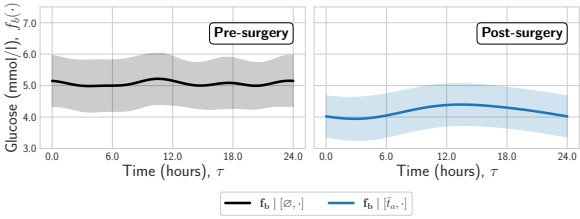

(a) Baseline progression.

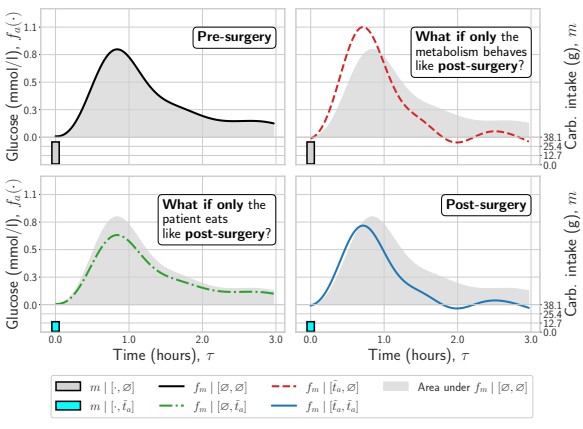

(b) Meal response functions.

Figure 4: Pre- vs. post-surgery predicted glucose progressions: **(a)** Baseline, **(b)** Meal response.

ing the diet in its natural pre-surgery state. Here, the meal response increases to a higher peak and declines back to the baseline faster than the shaded pre-surgery response, as previously described

[Jacobsen et al., 2013, Bojsen-Møller et al., 2014]. Moreover, the response model captures the decline under the baseline around $2\,h$ after a meal and a correction afterwards, which is a clinically-meaningful estimation suggesting overshooting of the glucose lowering due to the higher secretion of insulin after the surgery [Jacobsen et al., 2013, Bojsen-Møller et al., 2014]. In the **lower-left**, we predict the hypothetical meal response if patients *only* followed the post-surgery diet, while their metabolism would be fixed in its natural pre-surgery state. Here, the meal response increases to a lower peak due to a lower carbohydrate intake, while the shape of the response is preserved. In the **lower-right**, we predict a typical *post-surgery* response, which increases to a lower peak due to lower carbohydrate intake and declines back to the baseline faster compared to the pre-surgery response due to the high insulin levels.

### 5.1.3 Surgery Leads to Shorter Meal Intervals and Less Carbohydrate Intake

We compare meal data and predictions from the mediator model between the pre- and post-surgery periods in Fig. 5. The surgery leads to a more regular, low-carb diet due to the post-surgery dietary advice, a reduction in the stomach size, and an increase in appetite-regulating gut hormone levels [Dirksen et al., 2012, Laferrère and Pattou, 2018]. In Fig. 5a, we see that the surgery leads to a more peaky, unimodal distribution for the next meal time, compared to the multimodal pre-surgery distribution. The average time interval until the next meal decreases from $3.16\,h$ to $2.41\,h$. In Fig. 5b, we see that the post-surgery carbohydrate intake is more concentrated around the mean. The median per-meal carbohydrate intake decreases from $29.13\,g$ to $11.94\,g$.

### 5.1.4 Would intervening on the mediator (diet) suffice to regularize glucose progression?

As shown in Figs. 1, 4 and 5, the surgery affects glycemia through changing the diet (indirect) and other biological processes (direct). Then, a relevant research question is *how much of the surgery effect can be attributed to the changed diet*?

This question is crucial in assessing whether surgery affects blood glucose levels independently of dietary changes. If confirmed, it implies that surgery can influence blood glucose through two mechanisms: controlling meal size and independent hormonal changes. Additionally, surgery can also induce too low glucose levels (hypoglycemia), if the glucose peak after the meal induces too high insulin secretion [Lee et al., 2016].

To measure the contribution of direct and indirect causal paths, we use two metrics: (i) the percentage time spent in hypoglycemia (HG) $\%T_{\text{HG}} = \{t : y(t) \leq 3.9\,\text{mmol/l}\}/T$, and (ii) the percentage time spent in above-normal-glycemia (ANG) $\%T_{\text{ANG}} = \{t : y(t) \geq 5.6\,\text{mmol/l}\}/T$. We estimate direct, indirect and total effects on $\%T_{\text{HG}}$ and $\%T_{\text{ANG}}$, as in Eqs. (4) to (6). We show the results in Table 1. In the empirical data, $\%T_{\text{HG}}$ increased 27.5 points, while $\%T_{\text{ANG}}$ decreased 27.0 points. For the estimation of direct and indirect effects, we train our joint model on the RCT data set and use samples from the posterior to obtain a Monte Carlo approximation.

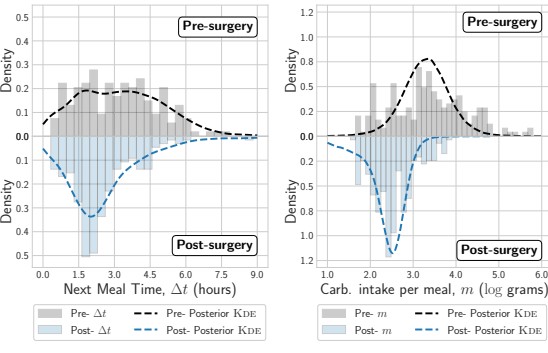

(a) Next meal arrival time.    (b) Carbohydrate intake.

Figure 5: Pre- vs. post-surgery data (histograms) and model posterior (kernel density estimate, dashed) for **(a)** next meal time and **(b)** (log) carbohydrate intake.

Table 1: NDE, NIE and TE results for % time spent in hypoglycemia $\%T_{\text{HG}}$ and in above-normal-glycemia $\%T_{\text{ANG}}$, to investigate *how much of the surgery effect can be attributed to a changed diet?*

| DATA TYPE | CAUSAL EFFECT | $\Delta\%T_{\text{HG}}$ | $\Delta\%T_{\text{ANG}}$ |
|---|---|---|---|
| Empirical data | TE | $+27.5$ | $-27.0$ |
| Simulation | TE | $+24.4 \pm 0.2$ | $-28.0 \pm 0.5$ |
| | NDE | $+24.7 \pm 0.2$ | $-26.4 \pm 0.5$ |
| | NIE | $-0.3 \pm 0.2$ | $-1.5 \pm 0.3$ |

The results suggest that the contribution of the changed diet (indirect, NIE) to the total effect in $\%T_{\text{HG}}$ and $\%T_{\text{ANG}}$ is much smaller compared to other metabolic processes (direct, NDE). Our computational exploration of these effects from observational data is supported empirically by Laferrère et al. [2008], who study the effects of two

Table 2: Mean squared error (MSE, mean $\pm$ std.dev. across 10 runs) for NDE, NIE and TE.

| JOINT MODEL | MODEL COMPONENTS | | | CAUSAL QUERY (MSE $\downarrow$) | | |
|---|---|---|---|---|---|---|
| | $A \rightarrow Y$ | MEDIATOR | RESPONSE | NDE($\tilde{t}_a$) | NIE($\tilde{t}_a$) | TE($\tilde{t}_a$) |
| M1 | ✓ | Non-interact. (L15) | Parametric (S17) | $0.22 \pm 0.02$ | $0.12 \pm 0.01$ | $0.31 \pm 0.02$ |
| M2 | ✓ | Non-interact. (L15) | Non-param. (H22) | $0.04 \pm 0.00$ | $0.15 \pm 0.01$ | $0.16 \pm 0.02$ |
| M3 | ✓ | Interacting (H22) | Parametric (S17) | $0.23 \pm 0.02$ | $0.11 \pm 0.01$ | $0.31 \pm 0.02$ |
| Z21-1 | ✓ | Interacting (H22) | Parametric (Z21) | $0.07 \pm 0.00$ | $0.09 \pm 0.01$ | $0.16 \pm 0.01$ |
| Z21-2 | ✓ | ORACLE | Parametric (Z21) | $0.07 \pm 0.00$ | $\mathbf{0.04 \pm 0.00}$ | $\mathbf{0.11 \pm 0.01}$ |
| H22 | ✗ | Interacting (H22) | Non-param. (H22) | – | – | $1.14 \pm 0.07$ |
| Our | ✓ | Interacting (H22) | Non-param. (H22) | $\mathbf{0.02 \pm 0.00}$ | $0.09 \pm 0.01$ | $\mathbf{0.10 \pm 0.01}$ |

different interventions — namely, dietary changes and surgical procedures — on individuals who had lost equal amounts of weight. In the study, the blood glucose response to a fixed carbohydrate meal shows a more pronounced decline in the surgery group compared to the control group. Because all patients received the same meal, this decline corresponds to the direct effect. This direct effect can be attributed to a range of intricate hormonal influences, including the involvement of gut hormones like GLP-1, alongside heightened insulin peaks subsequent to surgical intervention.

## 5.2 Semi-Synthetic Simulation Study

In this section, we set up a semi-synthetic simulation study to evaluate the proposed method on two causal tasks: estimating (i) natural direct effect (NDE) and (ii) natural indirect effect (NIE). For further details and additional results on sensitivity analysis, see Appendices C.2 and D.

**Simulator.** We train our joint model on the real-world RCT data set, and then use the learned model as the ground-truth meal–blood glucose simulator. One joint model is learned for each of the pre- and post-surgery periods. For the meal simulator, we use a mediator model learned on all patients. For the glucose simulator, we use an outcome model learned on a subset of three patients with distinct baselines and meal response functions to enable individualization between synthetic patients.

**Semi-synthetic dataset.** We use the ground-truth simulators to sample meal–glucose trajectories for $50$ synthetic patients. As in the RCT in Section 5.1, we assume each patient undergoes the surgery in the semi-synthetic scenario while being observed in pre- and post-surgery periods. For the training set, we sample $1$-day long trajectories for pre- and post-surgery periods. For the test set, we sample three interventional trajectories for each patient for the subsequent $1$-day, corresponding to three interventional distributions resulting from the following interventions: $\{[\varnothing, \varnothing], [\varnothing, \tilde{t}_a], [\tilde{t}_a, \tilde{t}_a]\}$. Then, the ground-truth NIE and NDE trajectories are calculated as in Eqs. (4) and (5), respectively.

**Benchmarks.** We compare methods with respect to their abilities in (i) allowing for an external intervention that jointly affects mediator and outcome processes, and (ii) capturing non-linear, long-range dependencies between these processes. For (i), we add a benchmark model [H22, Hızlı et al., 2023] that excludes the direct arrows from the treatment to the outcomes ($A \rightarrow Y$). For (ii), we combine different interacting/non-interacting mediator models and parametric/non-parametric response functions to obtain baseline methods that include the direct arrows ($A \rightarrow Y$). The methods Z21-1 and Z21-2 are named after a longitudinal causal mediation method [Zeng et al., 2021] with linear, instantaneous effects from mediator to outcome [Parametric (Z21), Zeng et al., 2021] (Fig. 2a). Additionally, we include a parametric response [Parametric (S17), Schulam and Saria, 2017] (M1, M3), and a non-parametric response [Non-parametric (H22), Hızlı et al., 2023] (M2). The non-interacting mediator model is chosen as a history-independent, non-homogeneous Poisson process [Non-interacting (L15), Lloyd et al., 2015] (M1, M2), while the interacting mediator model is chosen as a history-dependent, non-parametric point process model [Interacting (H22), Hızlı et al., 2023] (M3, Z21-1). In the joint model Z21-2, we further combine the parametric response model by Zeng et al. [2021] (Parametric (Z21)) with the ground-truth (ORACLE) mediator sequence.

**Metric.** On the test set, we report the mean squared error (MSE) between ground-truth and estimated trajectories. To sample comparable glucose trajectories, three interventional trajectories are simulated with fixed noise variables for meal (point process) sampling for all methods, as in Hızlı et al. [2023].

**Results.** MSE results are shown in Table 2. For the **natural direct effect**, formalized as $\text{NDE}(\tilde{t}_a) = Y_{>\tilde{t}_a}[\tilde{t}_a, \tilde{t}_a] - Y_{>\tilde{t}_a}[\varnothing, \tilde{t}_a]$, we answer the causal query: How differently will the glucose progress if we change *only* the metabolic processes other than the diet, while keeping the diet at its natural post-surgery value? We see that our joint model performs the best compared to the other methods. The models with the non-parametric response produce better NDE trajectories than the models with the parametric response, which fail to capture the glucose baseline and the meal response well. For the **natural indirect effect**, formalized as $\text{NIE}(\tilde{t}_a) = Y_{>\tilde{t}_a}[\varnothing, \tilde{t}_a] - Y_{>\tilde{t}_a}[\varnothing, \varnothing]$, we answer the causal query: How will the glucose progress if we change *only* the diet, while keeping the rest of the metabolic processes at their natural pre-surgery state? The oracle model Z21-2 that directly uses the mediator sequences sampled from the ground-truth simulator performs better than other methods. Our joint model performs better than the other methods that predict the mediator sequence, and it performs the same compared to the model Z21-1 which also uses the interacting mediator model. In contrast to the NDE results, the performance on the NIE improves with respect to the expressiveness of the mediator model, e.g. models with the interacting point process (M3, Z21-1, Our) perform better than the models with the non-interacting mediator (M1, M2). Similarly, the oracle model that has access to the ground-truth mediator sequences performs the best. For the **total effect**, formalized as $\text{TE}(\tilde{t}_a) = Y_{>\tilde{t}_a}[\tilde{t}_a, \tilde{t}_a] - Y_{\tilde{t}_a}[\varnothing, \varnothing]$, our joint model performs similar to the model Z21-2 that has access to the ground-truth mediators, while performing better than all the rest.

## 6 Limitations and Future Work

As required for causal inference, our identifiability result relies on causal assumptions, which may not hold in an observational study and are not statistically testable. For example, our assumption that there are no unobserved confounders between the mediator and outcome processes may not hold. To understand the sensitivity to this assumption, we performed an additional preliminary analysis in Appendix D, which can be refined in future work by extending the sensitivity analysis methods used in the static setting [Robins et al., 2000b]. In addition, our method relies on a hypothetical division of a treatment into path-specific treatments which solely affect the variables in their own path. This may not be possible in reality, e.g., it is not trivial to come up with a treatment that changes all the other metabolic processes without affecting the diet in our surgery–diet–blood glucose example.

In our real-world application, we focus on the dynamic causal mediation problem where the sequence of meals is the mediator of the effect of surgery on blood glucose. In this definition, the rest of the metabolic processes, e.g., gut hormones, weight loss, hepatic insulin sensitivity, or any other mediator through which the surgery could affect blood glucose, are included in the *direct effect*. An extension to multiple mediators and an explicit quantification of the corresponding path-specific effects will be an interesting future direction, especially from a medical point of view. In the semi-synthetic study, our joint model matches the functional class of the simulator; in future work, it would be valuable to further analyze the effect of model misspecification.

From a modeling perspective, the main limitation is scalability of the outcome model to a larger number of data points (see Appendix B.5 for details). This can be improved using separate sets of inducing points for the baseline function and the treatment response function. Alternatively, the GP models describing the components of our joint distribution (Eq. (8)) could be replaced by other model classes (e.g., neural networks). Our dynamic causal mediation framework does not put any fundamental limitations on the dimensionality of mediators or outcome.

## 7 Conclusion

We investigated temporal causal mediation in complex healthcare time-series by defining direct and indirect effects as stochastic processes instead of static random variables. Theoretically, we provided causal assumptions required for identifiability. To estimate the effects, we proposed a non-parametric mediator–outcome model that allows for an external intervention jointly affecting mediator and outcome sequences, while capturing time-delayed interactions between them. Despite the limitations, in a real-world study about the effect of bariatric surgery on meal–blood glucose dynamics, our method identified clinically-meaningful direct and indirect effects. This demonstrates how the model can be used for gaining insights from observational data without tedious or expensive experiments and supports its use for generating novel, testable research hypotheses. We believe that our method can lead to insightful analyses also in other domains such as epidemiology, public policy, and beyond.

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

# A  Causal Assumptions and Identifiability

## A.1  Static Causal Mediation

Consider a structural causal model (SCM) with a treatment $A$, a mediator $M$ and an outcome $Y$ (Fig. 6a). We define the total causal effect (TE) of the treatment $A$ on the outcome $Y$ as the difference between setting the treatment $A$ to a target treatment value $\tilde{a}$ and no treatment $\varnothing$ [Pearl, 2009]:

$$\text{TE}(\tilde{a}) = Y[\tilde{a}] - Y[\varnothing],$$

where we use the short-hand notation $[\tilde{a}]$ to denote an intervention: $[\tilde{a}] \equiv \text{do}(A = \tilde{a})$.

In many situations, the total effect does not capture the target of the scientific investigation. For instance, in the surgery–diet–glucose (*treatment–mediator–outcome*) example of Section 5.1, we examine how much of the surgery's effect is due to the post-surgery diet or the rest of the metabolic processes. This question can be formulated as a causal (mediation) query, where we estimate the indirect effect flowing through the mediator $M$ ($A{\to}M{\to}Y$), separately from the direct effect flowing through the path $A{\to}Y$. To measure the effect of a specific path, one can devise a pair of interventions on the treatment $A$ and the mediator $M$ so that only the specific target path is 'active'. A straightforward implementation of this idea leads to the definition of the controlled direct effect [CDE, Pearl, 2001]

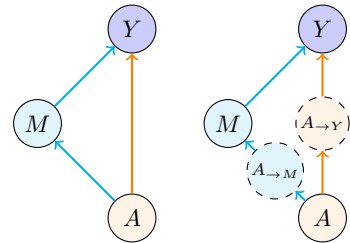

(a) Classical.     (b) Interventionist.

Figure 6: Graphical models for causal mediation. Mediation is defined in **(a)** using nested counterfactuals, while in **(b)** using path-specific treatments $A_{\to M}$ and $A_{\to Y}$.

$$\text{CDE}(\tilde{a}, \tilde{m}) = Y[A = \tilde{a}, M = \tilde{m}] - Y[A = \varnothing, M = \tilde{m}]$$
$$= Y[\tilde{a}, \tilde{m}] - Y[\varnothing, \tilde{m}], \tag{11}$$

where $[\tilde{a}, \tilde{m}]$ denotes a pair of interventions on $A$ and $M$: $[\tilde{a}, \tilde{m}] \equiv \text{do}(A = \tilde{a}, M = \tilde{m})$. The CDE is intuitively defined in Eq. (11) as the difference of the outcome $Y$ when the mediator $M$ is fixed to $\tilde{m}$ while the treatment $A$ is either $\tilde{a}$ or $\varnothing$. In the surgery–diet–glucose example, this corresponds to the difference in the outcome $Y$ while the patient is set to either (i) the surgery $\tilde{a}$ and a certain diet $\tilde{m}$ or (ii) no surgery $\varnothing$ and the same diet $\tilde{m}$. Hence, $\text{CDE}(\tilde{a}, \tilde{m})$ has a prescriptive interpretation as we prescribe both $A$ and $M$ to external intervention values, in contrast to the attributive interpretation that we are after to understand the underlying causal mechanisms [Pearl, 2001].

To understand the underlying causal mechanisms, we rather want to estimate the contribution of direct and indirect paths to the total causal effect. To achieve this, Pearl [2001] defines the natural direct effect (NDE) and the natural indirect effect (NIE). In natural effects, we devise a pair of interventions such that it sets a target path to a target intervention value while keeping the other path fixed in its natural state, i.e., the state that it would have attained without our external 'do'-intervention. Traditionally, the NDE and NIE are defined as a contrast between nested counterfactuals [Pearl, 2001]. Accordingly, the NDE is

$$\text{NDE}(\tilde{a}) = Y[A = \tilde{a}, M = \tilde{m}_{[A=\varnothing]}] - Y[A = \varnothing, M = \varnothing]$$
$$= Y[\tilde{a}, \tilde{m}_{[A=\varnothing]}] - Y[\varnothing, \varnothing], \tag{12}$$

where $[\tilde{a}, \tilde{m}_{[A=\varnothing]}] \equiv \text{do}(A = \tilde{a}, M = \tilde{m}_{\text{do}(A=\varnothing)})$ denotes a pair of interventions on $A$ and $M$, where the treatment $A$ is set to $\tilde{a}$ and the mediator $M$ is set to $\tilde{m}_{[A=\varnothing]}$, the value $M$ would have attained without our doing any intervention on the treatment $[A = \varnothing]$, i.e., no treatment. The counterfactual outcome $Y[\tilde{a}, \tilde{m}_{[A=\varnothing]}]$ is called 'nested' since we set the treatment to two distinct values $\tilde{a}$ and $\varnothing$ for the same individual data point $Y$, which is not possible in a real-world scenario. Similarly, the NIE is

$$\text{NIE}(\tilde{a}) = Y[A = \tilde{a}, M = \tilde{m}_{[A=\tilde{a}]}] - Y[A = \tilde{a}, M = \tilde{m}_{[A=\varnothing]}]$$
$$= Y[\tilde{a}, \tilde{m}_{[A=\tilde{a}]}] - Y[\tilde{a}, \tilde{m}_{[A=\varnothing]}]. \tag{13}$$

The NDE and NIE with nested counterfactuals are identifiable under the following causal assumptions [Pearl, 2001, 2014, Imai et al., 2010b]:

$\quad$ **(SA1):** $\quad p(\cdot \mid \text{do}(A = \tilde{a})) \equiv p(\cdot \mid A = \tilde{a}),$

$\quad$ **(SA2):** $\quad Y[\tilde{a}_1, \tilde{m}] \perp\!\!\!\perp M[\tilde{a}_2].$

The static assumption (**SA1**) states that there is no unobserved confounding (NUC) between the pairs $(A, M)$ and $(A, Y)$. Pearl [2014] interprets the static assumption (**SA2**) as the mediator–outcome relationship is deconfounded while keeping the treatment $A$ fixed. Under these assumptions, the distribution of the counterfactual outcome $Y[\tilde{a}, \tilde{m}_{[A=\varnothing]}]$ can be written as follows [Pearl, 2001]:

$$P(Y[\tilde{a}, \tilde{m}_{[A=\varnothing]}]) = \sum_{m^*} P(Y \mid A = \tilde{a}, M = m^*) P(M = m^* \mid A = \varnothing). \tag{14}$$

## A.2 Dynamic Causal Mediation (with a Single Intervention)

We extend the static definitions of natural direct and indirect effects by formulating them as stochastic processes over a period $[0, T]$. This leads to a definition of dynamic causal mediation. For each patient, we define (i) a treatment process $A$ that occurs only once at time $t_a \in [0, T]$, (ii) a mediator process $M$ that is observed at $I$ irregularly-sampled time points: $\{(t_i, m_i)\}_{i=1}^I$, and (iii) an outcome process $Y$ that is observed at $J$ irregularly-sampled time points: $\{(t_j, y_j)\}_{j=1}^J$. For a detailed problem formulation, see Section 2.1.

The identifiability in the nested-counterfactual based mediation is *only* possible if there exist no *post-treatment confounders*, i.e., confounder variables between a mediator and an outcome must not be causally affected by the treatment [Pearl, 2014]. However, in dynamic causal mediation, past mediators and outcomes causally affect future mediators and outcomes. This means that a past outcome measurement acts as a post-treatment confounder for a future mediator and a future outcome, and the NDE and NIE are non-identifiable without further assumptions [Pearl, 2014, Didelez, 2019].

To work around the post-treatment confounding problem, Zeng et al. [2021] assume that there exists no interaction from past outcomes to future mediators (See Fig. 2a). However, in a real-world healthcare setup, the mediator and outcome processes most likely interact, e.g., past glucose levels should affect future glucose levels as well as future meals. To overcome this limitation, we follow an interventionist approach to the causal mediation [Robins et al., 2022], similar to Didelez [2019], Aalen et al. [2020]. In this approach, we hypothetically duplicate the treatment $A$ into two path-specific treatments: the direct-path treatment $A_{\to Y}$ and the indirect-path treatment $A_{\to M}$ (Fig. 6b). For instance, in the surgery–diet–glucose example, the direct-path treatment $A_{\to Y}$ corresponds to changing *only* the metabolic processes (other than the diet), without affecting the diet. Using path-specific interventions $A_{\to M}$ and $A_{\to Y}$, we define the NDE as

$$\text{NDE}(\tilde{t}_a) \equiv \boldsymbol{Y}_{>\tilde{t}_a}[A_{\to Y} = \tilde{t}_a, A_{\to M} = \tilde{t}_a] - \boldsymbol{Y}_{>\tilde{t}_a}[A_{\to Y} = \varnothing, A_{\to M} = \tilde{t}_a]$$
$$\equiv \boldsymbol{Y}_{>\tilde{t}_a}[\tilde{t}_a, \tilde{t}_a] - \boldsymbol{Y}_{>\tilde{t}_a}[\varnothing, \tilde{t}_a], \tag{15}$$

where $Y[\tilde{a}_1, \tilde{a}_2]$ is the *potential* outcome after setting the direct-path treatment $A_{\to Y}$ to $\tilde{a}_1$ and the indirect-path treatment $A_{\to M}$ to $\tilde{a}_2$. Similarly, the NIE is defined as

$$\text{NIE}(\tilde{t}_a) \equiv \boldsymbol{Y}_{>\tilde{t}_a}[A_{\to Y} = \varnothing, A_{\to M} = \tilde{t}_a] - \boldsymbol{Y}_{>\tilde{t}_a}[A_{\to Y} = \varnothing, A_{\to M} = \varnothing]$$
$$\equiv \boldsymbol{Y}_{>\tilde{t}_a}[\varnothing, \tilde{t}_a] - \boldsymbol{Y}_{>\tilde{t}_a}[\varnothing, \varnothing]. \tag{16}$$

Accordingly, we define the total effect as follows

$$\text{TE}(\tilde{t}_a) = \text{NDE}(\tilde{t}_a) + \text{NIE}(\tilde{t}_a) = \boldsymbol{Y}_{>\tilde{t}_a}[\tilde{t}_a, \tilde{t}_a] - \boldsymbol{Y}_{>\tilde{t}_a}[\varnothing, \varnothing]. \tag{17}$$

### A.2.1 Causal Assumptions

To identify the NDE and the NIE, we make the causal assumptions (**A1**, **A2**, **A3**):

**Assumption A1: Continuous-time NUC.** There are no unobserved confounders (NUC) between the treatment–mediator process pair $(A, M)$ and the treatment–outcome process pair $(A, Y)$, i.e., the treatment is randomized. Hence, the interventional distribution of a treatment intervention and the conditional distribution of those who got treated are equivalent:

$$p(\cdot \mid \text{do}(\boldsymbol{A} = \tilde{\boldsymbol{N}}_A)) \equiv p(\cdot \mid \boldsymbol{A} = \tilde{\boldsymbol{N}}_A). \tag{18}$$

**Assumption A2: Local Independence of Mediator $M$ and a Single Direct-Path Treatment $A_{\to Y}$.** Conditioned on the history $\mathcal{H}_{<\tau}$ of the mediator and outcome processes up until time $\tau$, the mediator process $M$ is independent of the direct-path treatment $A_{\to Y}$ locally at time $\tau$, for all times $\tau \in [0, T]$:

$$\boldsymbol{A}_{\to Y} \perp\!\!\!\perp M(\tau) \mid \boldsymbol{A}_{\to M}, \mathcal{H}_{<\tau}, \quad \tau \in [0, T]. \tag{19}$$

Hence, there are no direct arrows from the direct-path treatment $A_{\to Y}$ to the mediator process $M$.

**Assumption A3: Local Independence of Outcome $Y$ and a Single Indirect-Path Treatment $\boldsymbol{A}_{\to M}$.** Conditioned on the history $\mathcal{H}_{<\tau}$ of the mediator and outcome processes up until time $\tau$, the outcome process $Y(\tau)$ is independent of the indirect-path treatment $\boldsymbol{A}_{\to M}$ locally at time $\tau$, for all time $\tau \in [0, T]$:

$$\boldsymbol{A}_{\to M} \perp\!\!\!\perp Y(\tau) \mid \boldsymbol{A}_{\to Y}, \mathcal{H}_{<\tau}, \quad \tau \in [0, T]. \tag{20}$$

Hence, there are no direct arrows from the indirect-path treatment $\boldsymbol{A}_{\to M}$ to the outcome process $Y$.

The static version of the NUC assumption in (**A1**) is common in the existing literature on longitudinal causal mediation [Didelez, 2019, Aalen et al., 2020]. Here, we define it as a continuous-time NUC [Schulam and Saria, 2017] for completeness, as we also define the treatment $A$ as a process in Section 2.1. (**A2, A3**) state that the path-specific treatments $\boldsymbol{A}_{\to Y}$ and $\boldsymbol{A}_{\to M}$ causally affect only their target path for a given time $\tau$ conditioned on the past history. They can be seen as the continuous-time generalizations of the similar causal assumptions provided for discrete-time mediator–outcome measurements $(M_1, Y_1, M_2, Y_2, \ldots, M_n, Y_n)$ in Didelez [2019] and Aalen et al. [2020]. An interesting connection is between (**A2, A3**) and the notion of local independence [Didelez, 2006, 2008, 2015], a dynamic interpretation of independence that is the continuous-time version of Granger causality [Granger, 1969]. In this regard, (**A2, A3**) state that the processes $M$ and $Y$ are locally independent of the direct- and indirect-path treatments $\boldsymbol{A}_{\to Y}$ and $\boldsymbol{A}_{\to M}$ respectively, given the history of the rest of the processes. Furthermore, (**A2, A3**) imply that there are no (continuous-time) unobserved confounders between the mediator and outcome processes, otherwise an unobserved confounder between them would render the process $Y$ (or $M$) and $\boldsymbol{A}_{\to M}$ (or $\boldsymbol{A}_{\to Y}$) dependent as we condition on the past history (See Didelez [2019] for a detailed explanation).

### A.2.2 Proof of the Identifiability Result

For the total effect, we need to estimate the trajectories $\boldsymbol{Y}_{>\tilde{t}_a}[\tilde{t}_a, \tilde{t}_a]$ and $\boldsymbol{Y}_{>\tilde{t}_a}[\varnothing, \varnothing]$, which are identifiable under (**A1**). For direct and indirect effects, we further need to estimate the counterfactual trajectory $\boldsymbol{Y}_{>\tilde{t}_a}[\varnothing, \tilde{t}_a]$, i.e., the outcome process under a hypothetical intervention, which is identifiable under (**A1, A2, A3**). In the following, we show the identification result for the outcome process under a hypothetical intervention $\boldsymbol{Y}_{>\tilde{t}_a}[\varnothing, \tilde{t}_a]$, since it is a generalization of the 'factual' trajectories $\boldsymbol{Y}_{>\tilde{t}_a}[\tilde{t}_a, \tilde{t}_a]$ and $\boldsymbol{Y}_{>\tilde{t}_a}[\varnothing, \varnothing]$.

We consider a counterfactual trajectory $\boldsymbol{Y}_{>\tilde{t}_a}$ under a paired intervention, e.g. $[\varnothing, \tilde{t}_a]$, at $R$ ordered query points $\mathbf{q} = \{q_r\}_{r=1}^R$, $q_r > \tilde{t}_a$: $\boldsymbol{Y}_{\mathbf{q}}[\varnothing, \tilde{t}_a] = \{Y(q_r)[\varnothing, \tilde{t}_a]\}_{r=1}^R$. We start by explicitly writing the interventional trajectory in do-notation:

$$P(\boldsymbol{Y}_{\mathbf{q}}[\varnothing, \tilde{t}_a]) = P(\boldsymbol{Y}_{\mathbf{q}} \mid \mathrm{do}(t_{A_{\to Y}} = \varnothing, t_{A_{\to M}} = \tilde{t}_a)).$$

Let $M_{[q_r, q_{r+1})}$ denote the mediator occurrences between two consecutive query points $q_r$ and $q_{r+1}$ and $\boldsymbol{M}_{>\tilde{t}_a} = \cup_{r=0}^{R-1} M_{[q_r, q_{r+1})}$. We include the mediator process $\boldsymbol{M}_{>\tilde{t}_a}$ to the counterfactual trajectory $\boldsymbol{Y}_{\mathbf{q}}[\varnothing, \tilde{t}_a]$, and write the conditional distributions of $\boldsymbol{Y}_{\mathbf{q}}$ and $\boldsymbol{M}_{>\tilde{t}_a}$ using a factorization in temporal order:

$$P(\boldsymbol{Y}_{\mathbf{q}}[\varnothing, \tilde{t}_a]) = \sum_{\boldsymbol{M}_{>\tilde{t}_a}} \prod_{r=0}^{R-1} P(Y_{q_{r+1}} \mid \mathrm{do}(t_{A_{\to Y}} = \varnothing, t_{A_{\to M}} = \tilde{t}_a), M_{[q_r, q_{r+1})}, \mathcal{H}_{\leq q_r})$$

$$P(M_{[q_r, q_{r+1})} \mid \mathrm{do}(t_{A_{\to Y}} = \varnothing, t_{A_{\to M}} = \tilde{t}_a), \mathcal{H}_{\leq q_r}), \tag{21}$$

where $q_0 = \tilde{t}_a$ and $\mathcal{H}_{\leq q_r}$ denotes the history that contains the past information on the path-specific treatments $(\boldsymbol{A}_{\to M}, \boldsymbol{A}_{\to Y})$ and past mediators up to (non-inclusive) $q_r$, and past outcomes up until (inclusive) $q_r$: $\mathcal{H}_{\leq q_r} = \boldsymbol{A}_{\to M} \cup \boldsymbol{A}_{\to Y} \cup \{(t_i, m_i) : t_i < q_r\} \cup \{(t_j, y_j) : t_j \leq q_r\}$.

(**A3**) states that the outcome $Y(q_{r+1})$ is independent of the indirect-path treatment $A_{\to M}$ conditioned on the past history $\mathcal{H}_{<q_{r+1}}$ including the direct-path treatment $A_{\to Y}$. Hence, we can change the value of the intervention on $t_{A_{\to M}}$ without changing the first conditional distribution term in Eq. (21). Using (**A3**), we re-write Eq. (21) as

$$P(\boldsymbol{Y}_{\mathbf{q}}[\varnothing, \tilde{t}_a]) = \sum_{\boldsymbol{M}_{>\tilde{t}_a}} \prod_{r=0}^{R-1} P(Y_{q_{r+1}} \mid \mathrm{do}(t_{A_{\to Y}} = \varnothing, t_{A_{\to M}} = \varnothing), M_{[q_r, q_{r+1})}, \mathcal{H}_{\leq q_r})$$

$$P(M_{[q_r, q_{r+1})} \mid \mathrm{do}(t_{A_{\to Y}} = \varnothing, t_{A_{\to M}} = \tilde{t}_a), \mathcal{H}_{\leq q_r}). \tag{22}$$

Similarly, (**A2**) states that the mediators $M_{[q_r, q_{r+1})}$ are independent of the direct-path treatment $A_{\to Y}$ conditioned on the past history $\mathcal{H}_{<q_r}$ including the indirect-path treatment $A_{\to M}$. Using (**A2**), we re-write Eq. (22) as

$$P(\mathbf{Y_q}[\varnothing, \tilde{t}_a]) = \sum_{\mathbf{M}_{>\tilde{t}_a}} \prod_{r=0}^{R-1} P(Y_{q_{r+1}} | \operatorname{do}(t_{A_{\to Y}} = \varnothing, t_{A_{\to M}} = \varnothing), M_{[q_r, q_{r+1})}, \mathcal{H}_{\leq q_r})$$
$$P(M_{[q_r, q_{r+1})} | \operatorname{do}(t_{A_{\to Y}} = \tilde{t}_a, t_{A_{\to M}} = \tilde{t}_a), \mathcal{H}_{\leq q_r}). \tag{23}$$

In Eq. (23), the target quantities are considered under 'factual' intervention pairs:

$$\operatorname{do}(t_{A_{\to Y}} = \varnothing, t_{A_{\to M}} = \varnothing) \equiv \operatorname{do}(t_A = \varnothing), \tag{24}$$
$$\operatorname{do}(t_{A_{\to Y}} = \tilde{t}_a, t_{A_{\to M}} = \tilde{t}_a) \equiv \operatorname{do}(t_A = \tilde{t}_a). \tag{25}$$

Using Eqs. (24) and (25), we can simplify Eq. (23) as follows:

$$P(\mathbf{Y_q}[\varnothing, \tilde{t}_a]) = \sum_{\mathbf{M}_{>\tilde{t}_a}} \prod_{r=0}^{R-1} P(Y_{q_{r+1}} | \operatorname{do}(t_A = \varnothing), M_{[q_r, q_{r+1})}, \mathcal{H}_{\leq q_r})$$
$$P(M_{[q_r, q_{r+1})} | \operatorname{do}(t_A = \tilde{t}_a), \mathcal{H}_{\leq q_r}). \tag{26}$$

In Eq. (26), both terms in the product correspond to the mediator and outcome processes under the interventions $[\varnothing]$ and $[\tilde{t}_a]$ respectively. These interventional distributions are identified under (**A1**), as it states that there are no unobserved confounders between the process pairs $(\mathbf{A}, \mathbf{M})$ and $(\mathbf{A}, \mathbf{Y})$. Using (**A1**) on Eq. (26), we obtain the final identification result:

$$P(\mathbf{Y_q}[\varnothing, \tilde{t}_a]) = \sum_{\mathbf{M}_{>\tilde{t}_a}} \prod_{r=0}^{R-1} \underbrace{P(Y_{q_{r+1}} | t_A = \varnothing, M_{[q_r, q_{r+1})}, \mathcal{H}_{\leq q_r})}_{\text{outcome term } \bullet} \underbrace{P(M_{[q_r, q_{r+1})} | t_A = \tilde{t}_a, \mathcal{H}_{\leq q_r})}_{\text{mediator intensity } \circ}.$$
$$\tag{27}$$

To estimate the direct and indirect effects, we model the two terms in Eq. (27) with an interacting mediator–outcome model.

## A.3 Dynamic Causal Mediation (Generalization to a Treatment Sequence)

In this section, we extend the dynamic causal mediation definitions to multiple treatments. To achieve this, we define the treatment as a non-binary, non-unitary process $\mathbf{A} : [0, T] \to \mathbb{N} \times \{\mathbb{R} \cup \{\varnothing\}\}$ similar to the mediator process $\mathbf{M}$. Here, the difference between two processes $\mathbf{A}$ and $\mathbf{M}$ is that we consider $\mathbf{A}$ to be a known, deterministic sequence, while $\mathbf{M}$ to be unknown and stochastic. Hence, we assume that the treatment process $\mathbf{A}$ is given as a sequence $\mathbf{a} = \{a_n\}_{n=1}^N$, where each treatment $a_n = (t_n, d_n)$ is a tuple of a treatment time $t_n \in [0, T]$ and a treatment type/dosage marker $d_n \in \mathcal{A}$. The type/dosage space can be chosen as a discrete space $\mathcal{A} = \mathbb{Z}$ representing a set of treatment types, a continuous space $\mathcal{A} = \mathbb{R}$ representing a set of dosages, or a combination of both spaces $\mathcal{A} = \mathbb{N} \times \mathbb{R}$.

We again hypothetically duplicate the treatment process $\mathbf{A}$ into two path-specific treatment sequences: the direct-path treatments $\mathbf{A}_{\to Y}$ and the indirect-path treatments $\mathbf{A}_{\to M}$ (Fig. 6b). Using path-specific interventions $\mathbf{A}_{\to M}$ and $\mathbf{A}_{\to Y}$, we define the NDE as

$$\text{NDE}(\tilde{\mathbf{a}}_1, \tilde{\mathbf{a}}_0) \equiv \mathbf{Y}_{>\tilde{t}_a}[\mathbf{A}_{\to Y} = \tilde{\mathbf{a}}_1, \mathbf{A}_{\to M} = \tilde{\mathbf{a}}_1] - \mathbf{Y}_{>\tilde{t}_a}[\mathbf{A}_{\to Y} = \tilde{\mathbf{a}}_0, \mathbf{A}_{\to M} = \tilde{\mathbf{a}}_1]$$
$$\equiv \mathbf{Y}_{>\tilde{t}_a}[\tilde{\mathbf{a}}_1, \tilde{\mathbf{a}}_1] - \mathbf{Y}_{>\tilde{t}_a}[\tilde{\mathbf{a}}_0, \tilde{\mathbf{a}}_1], \tag{28}$$

where $Y[\tilde{\mathbf{a}}_0, \tilde{\mathbf{a}}_1]$ is the *potential* outcome after setting the direct-path treatment sequence $\mathbf{A}_{\to Y}$ to $\tilde{\mathbf{a}}_0 = \{a_{0,1}, a_{0,2}, \ldots, a_{0,N_0}\}$ and the indirect-path treatment sequence $\mathbf{A}_{\to M}$ to $\tilde{\mathbf{a}}_1 = \{a_{1,1}, a_{1,2}, \ldots, a_{1,N_1}\}$. Similarly, the NIE is defined as

$$\text{NIE}(\tilde{\mathbf{a}}_1, \tilde{\mathbf{a}}_0) \equiv \mathbf{Y}_{>\tilde{t}_a}[\mathbf{A}_{\to Y} = \tilde{\mathbf{a}}_0, \mathbf{A}_{\to M} = \tilde{\mathbf{a}}_1] - \mathbf{Y}_{>\tilde{t}_a}[\mathbf{A}_{\to Y} = \tilde{\mathbf{a}}_0, \mathbf{A}_{\to M} = \tilde{\mathbf{a}}_0]$$
$$\equiv \mathbf{Y}_{>\tilde{t}_a}[\tilde{\mathbf{a}}_0, \tilde{\mathbf{a}}_1] - \mathbf{Y}_{>\tilde{t}_a}[\tilde{\mathbf{a}}_0, \tilde{\mathbf{a}}_0]. \tag{29}$$

Accordingly, we define the total effect as follows

$$\text{TE}(\tilde{\mathbf{a}}_1, \tilde{\mathbf{a}}_0) = \text{NDE}(\tilde{\mathbf{a}}_1, \tilde{\mathbf{a}}_0) + \text{NIE}(\tilde{\mathbf{a}}_1, \tilde{\mathbf{a}}_0) = \mathbf{Y}_{>\tilde{t}_a}[\tilde{\mathbf{a}}_1, \tilde{\mathbf{a}}_1] - \mathbf{Y}_{>\tilde{t}_a}[\tilde{\mathbf{a}}_0, \tilde{\mathbf{a}}_0]. \tag{30}$$

### A.3.1 Causal Assumptions

To identify the NDE and the NIE, we make the causal assumptions (**A4**, **A5**, **A6**):

**Assumption A4: Continuous-time NUC [Hızlı et al., 2023].** Let sequential treatments **a** occur at a discrete set of time points on a given interval, characterized by the conditional intensity function $\lambda_a^*(t, d)$. The conditional treatment intensity $\lambda_a^*(t, d)$ of a treatment $a = (t, d)$, is independent of the potential outcome trajectory $\boldsymbol{Y}_{>\tau}[\tilde{\mathbf{a}}]$, conditioned on the past history $\mathcal{H}_{<\tau}, \forall t \in \mathbb{R}_{\geq 0}$, where the history $\mathcal{H}_{<\tau}$ contains the past treatments, mediators and outcomes.

**Assumption A5: Local Independence of Mediator $M$ and Direct-Path Treatment $A_{\rightarrow Y}$.** Conditioned on the indirect-path treatment process $\boldsymbol{A}_{<\tau}^{\rightarrow M}$ and the history $\mathcal{H}_{<\tau}$ of the mediator and outcome processes up until time $\tau$, the mediator process $M$ is independent of the history of the direct-path treatment $\boldsymbol{A}_{<\tau}^{\rightarrow Y}$ locally at time $\tau$, for all times $\tau \in [0, T]$:

$$\boldsymbol{A}_{<\tau}^{\rightarrow Y} \perp\!\!\!\perp M(\tau) \mid \boldsymbol{A}_{<\tau}^{\rightarrow M}, \mathcal{H}_{<\tau}, \quad \tau \in [0, T]. \tag{31}$$

Hence, there are no direct arrows from the direct-path treatment $\boldsymbol{A}_{\rightarrow Y}$ to the mediator process $M$.

**Assumption A6: Local Independence of Outcome $Y$ and Indirect-Path Treatment $A_{\rightarrow M}$.** Conditioned on the indirect-path treatment process $\boldsymbol{A}_{<\tau}^{\rightarrow Y}$ and the history $\mathcal{H}_{<\tau}$ of the mediator and outcome processes up until time $\tau$, the outcome process $Y(\tau)$ is independent of the past of the indirect-path treatment $\boldsymbol{A}_{<\tau}^{\rightarrow M}$ locally at time $\tau$, for all time $\tau \in [0, T]$:

$$\boldsymbol{A}_{<\tau}^{\rightarrow M} \perp\!\!\!\perp Y(\tau) \mid \boldsymbol{A}_{<\tau}^{\rightarrow Y}, \mathcal{H}_{<\tau}, \quad \tau \in [0, T]. \tag{32}$$

Hence, there are no direct arrows from the indirect-path treatment $\boldsymbol{A}_{\rightarrow M}$ to the outcome process $Y$.

Assumptions (**A5**, **A6**) define the local independence with respect to the past sequences of the direct- and indirect-path treatment processes $\boldsymbol{A}_{<\tau}^{\rightarrow Y}$ and $\boldsymbol{A}_{<\tau}^{\rightarrow M}$. This is in contrast with the definitions of Assumptions (**A2**, **A3**), where we only have a single treatment.

### A.3.2 Proof of the Identifiability Result

For the total effect, we need to estimate the trajectories $\boldsymbol{Y}_{>\tilde{t}_a}[\tilde{\mathbf{a}}_1, \tilde{\mathbf{a}}_1]$ and $\boldsymbol{Y}_{>\tilde{t}_a}[\tilde{\mathbf{a}}_0, \tilde{\mathbf{a}}_0]$, which are identifiable under (**A4**). For direct and indirect effects, we further need to estimate the counterfactual trajectory $\boldsymbol{Y}_{>\tilde{t}_a}[\tilde{\mathbf{a}}_0, \tilde{\mathbf{a}}_1]$, i.e., the outcome process under a hypothetical intervention, which is identifiable under (**A4**, **A5**, **A6**). In the following, we show the identification result for the outcome process under a hypothetical intervention $\boldsymbol{Y}_{>\tilde{t}_a}[\tilde{\mathbf{a}}_0, \tilde{\mathbf{a}}_1]$, since it is a generalization of the 'factual' trajectories $\boldsymbol{Y}_{>\tilde{t}_a}[\tilde{\mathbf{a}}_1, \tilde{\mathbf{a}}_1]$ and $\boldsymbol{Y}_{>\tilde{t}_a}[\tilde{\mathbf{a}}_0, \tilde{\mathbf{a}}_0]$.

We consider a counterfactual trajectory $\boldsymbol{Y}_{>\tilde{t}_a}$ under a paired intervention, e.g. $[\tilde{\mathbf{a}}_0, \tilde{\mathbf{a}}_1]$, at $R$ ordered query points $\mathbf{q} = \{q_r\}_{r=1}^R, q_r > \tilde{t}_a$: $\boldsymbol{Y}_{\mathbf{q}}[\tilde{\mathbf{a}}_0, \tilde{\mathbf{a}}_1] = \{Y(q_r)[\tilde{\mathbf{a}}_0, \tilde{\mathbf{a}}_1]\}_{r=1}^R$. We again start by explicitly writing the interventional trajectory in do-notation:

$$P(\boldsymbol{Y}_{\mathbf{q}}[\varnothing, \tilde{t}_a]) = P(\boldsymbol{Y}_{\mathbf{q}} \mid \text{do}(\boldsymbol{A}^{\rightarrow Y} = \tilde{\mathbf{a}}_0, \boldsymbol{A}^{\rightarrow M} = \tilde{\mathbf{a}}_1)).$$

Similar to the analysis with a single intervention in Appendix A.2.2, we define $M_{>\tilde{t}_a} = \cup_{r=0}^{R-1} M_{[q_r, q_{r+1})}$ and write the conditional distributions of $\boldsymbol{Y}_{\mathbf{q}}$ and $M_{>\tilde{t}_a}$ using a factorization in temporal order:

$$P(\boldsymbol{Y}_{\mathbf{q}}[\tilde{\mathbf{a}}_0, \tilde{\mathbf{a}}_1]) = \sum_{M_{>\tilde{t}_a}} \prod_{r=0}^{R-1} P(Y_{q_{r+1}} \mid \text{do}(\boldsymbol{A}^{\rightarrow Y} = \tilde{\mathbf{a}}_0, \boldsymbol{A}^{\rightarrow M} = \tilde{\mathbf{a}}_1), M_{[q_r, q_{r+1})}, \mathcal{H}_{\leq q_r})$$

$$P(M_{[q_r, q_{r+1})} \mid \text{do}(\boldsymbol{A}^{\rightarrow Y} = \tilde{\mathbf{a}}_0, \boldsymbol{A}^{\rightarrow M} = \tilde{\mathbf{a}}_1), \mathcal{H}_{\leq q_r}), \tag{33}$$

where $q_0 = \tilde{t}_a$ and $\mathcal{H}_{\leq q_r}$ denotes the history that contains the past information on the path-specific treatments $(\boldsymbol{A}^{\rightarrow M}, \boldsymbol{A}^{\rightarrow Y})$ and past mediators up to (non-inclusive) $q_r$, and past outcomes up until (inclusive) $q_r$: $\mathcal{H}_{\leq q_r} = \boldsymbol{A}_{\leq q_r}^{\rightarrow M} \cup \boldsymbol{A}_{\leq q_r}^{\rightarrow Y} \cup \{(t_i, m_i) : t_i < q_r\} \cup \{(t_j, y_j) : t_j \leq q_r\}$.

Let us start with the first term $P(Y_{q_{r+1}} \mid \text{do}(\boldsymbol{A}^{\rightarrow Y} = \tilde{\mathbf{a}}_0, \boldsymbol{A}^{\rightarrow M} = \tilde{\mathbf{a}}_1), M_{[q_r, q_{r+1})}, \mathcal{H}_{\leq q_r})$. We know that the outcome $Y_{q_{r+1}}$ at time point $q_{r+1}$ is independent of any future interventions:

$$P(Y_{q_{r+1}} \mid \text{do}(\boldsymbol{A}^{\rightarrow Y} = \tilde{\mathbf{a}}_0, \boldsymbol{A}^{\rightarrow M} = \tilde{\mathbf{a}}_1), M_{[q_r, q_{r+1})}, \mathcal{H}_{\leq q_r}) =$$
$$= P(Y_{q_{r+1}} \mid \text{do}(\boldsymbol{A}_{<q_{r+1}}^{\rightarrow Y} = \tilde{\mathbf{a}}_{0, <q_{r+1}}, \boldsymbol{A}_{<q_{r+1}}^{\rightarrow M} = \tilde{\mathbf{a}}_{1, <q_{r+1}}), M_{[q_r, q_{r+1})}, \mathcal{H}_{\leq q_r}). \tag{34}$$

(**A6**) now states that the outcome $Y(q_{r+1})$ is independent of the past of the indirect-path treatment $\boldsymbol{A}^{\to M}_{<q_{r+1}}$ conditioned on the past history $\mathcal{H}_{<q_{r+1}}$ including the past of the direct-path treatment $\boldsymbol{A}^{\to Y}_{<q_{r+1}}$. Hence, we can change the value of the intervention on $\boldsymbol{A}^{\to M}_{<q_{r+1}}$ without changing probability of the conditional term in Eq. (34). Using (**A6**), we re-write Eq. (34) as

$$P(Y_{q_{r+1}}|\operatorname{do}(\boldsymbol{A}^{\to Y}_{<q_{r+1}} = \tilde{\mathbf{a}}_{0,<q_{r+1}}, \boldsymbol{A}^{\to M}_{<q_{r+1}} = \tilde{\mathbf{a}}_{1,<q_{r+1}}), M_{[q_r,q_{r+1})}, \mathcal{H}_{\leq q_r})$$
$$= P(Y_{q_{r+1}}|\operatorname{do}(\boldsymbol{A}^{\to Y}_{<q_{r+1}} = \tilde{\mathbf{a}}_{0,<q_{r+1}}, \boldsymbol{A}^{\to M}_{<q_{r+1}} = \tilde{\mathbf{a}}_{0,<q_{r+1}}), M_{[q_r,q_{r+1})}, \mathcal{H}_{\leq q_r}). \quad (35)$$

Let us continue with the mediator term $P(M_{[q_r,q_{r+1})}|\operatorname{do}(\boldsymbol{A}^{\to Y} = \tilde{\mathbf{a}}_0, \boldsymbol{A}^{\to M} = \tilde{\mathbf{a}}_1), \mathcal{H}_{\leq q_r})$. The steps will be similar to the discussion above, with one difference: there might be interventions $\tilde{\mathbf{a}}_{[q_r,q_{r+1})}$ in the interval $[q_r, q_{r+1})$. Without loss of generality, let us assume there is only a single intervention $a_p = (t_p, d_p)$ in the interval $t_p \in [q_r, q_{r+1})$. Then, we consider the mediator term in two parts, namely up to $t_p$ and after $t_p$:

$$P(M_{[q_r,q_{r+1})}|\operatorname{do}(\boldsymbol{A}^{\to Y} = \tilde{\mathbf{a}}_0, \boldsymbol{A}^{\to M} = \tilde{\mathbf{a}}_1), \mathcal{H}_{\leq q_r})$$
$$= P(M_{[q_r,t_p]}|\operatorname{do}(\boldsymbol{A}^{\to Y} = \tilde{\mathbf{a}}_0, \boldsymbol{A}^{\to M} = \tilde{\mathbf{a}}_1), \mathcal{H}_{\leq q_r})$$
$$P(M_{(t_p,q_{r+1})}|\operatorname{do}(\boldsymbol{A}^{\to Y} = \tilde{\mathbf{a}}_0, \boldsymbol{A}^{\to M} = \tilde{\mathbf{a}}_1), M_{[q_r,t_p]}, \mathcal{H}_{\leq q_r}) \quad (36)$$

We know that the mediators $M_{[q_r,t_p]}$ and $M_{(t_p,q_{r+1})}$ are independent of any future interventions:

$$P(M_{[q_r,t_p]}|\operatorname{do}(\boldsymbol{A}^{\to Y} = \tilde{\mathbf{a}}_0, \boldsymbol{A}^{\to M} = \tilde{\mathbf{a}}_1), \mathcal{H}_{\leq q_r})$$
$$= P(M_{[q_r,t_p]}|\operatorname{do}(\boldsymbol{A}^{\to Y}_{<q_r} = \tilde{\mathbf{a}}_{0,<q_r}, \boldsymbol{A}^{\to M}_{<q_r} = \tilde{\mathbf{a}}_{1,<q_r}), \mathcal{H}_{\leq q_r}) \quad (37)$$

$$P(M_{(t_p,q_{r+1})}|\operatorname{do}(\boldsymbol{A}^{\to Y} = \tilde{\mathbf{a}}_0, \boldsymbol{A}^{\to M} = \tilde{\mathbf{a}}_1), M_{[q_r,t_p]}, \mathcal{H}_{\leq q_r})$$
$$= P(M_{(t_p,q_{r+1})}|\operatorname{do}(\boldsymbol{A}^{\to Y}_{\leq t_p} = \tilde{\mathbf{a}}_{0,\leq t_p}, \boldsymbol{A}^{\to M}_{\leq t_p} = \tilde{\mathbf{a}}_{1,\leq t_p}), M_{[q_r,t_p]}, \mathcal{H}_{\leq q_r}) \quad (38)$$

(**A5**) states that the mediator $M_{[q_r,t_p]}$ (and $M_{(t_p,q_{r+1})}$) is independent of the direct-path treatment $\boldsymbol{A}^{\to Y}_{<q_r}$ conditioned on the past history $\mathcal{H}_{<q_r}$ including the indirect-path treatment $\boldsymbol{A}^{\to M}_{<q_r}$. Using (**A5**), we re-write Eq. (37) and Eq. (38) as

$$P(M_{[q_r,t_p]}|\operatorname{do}(\boldsymbol{A}^{\to Y}_{<q_r} = \tilde{\mathbf{a}}_{0,<q_r}, \boldsymbol{A}^{\to M}_{<q_r} = \tilde{\mathbf{a}}_{1,<q_r}), \mathcal{H}_{\leq q_r})$$
$$= P(M_{[q_r,t_p]}|\operatorname{do}(\boldsymbol{A}^{\to Y}_{<q_r} = \tilde{\mathbf{a}}_{1,<q_r}, \boldsymbol{A}^{\to M}_{<q_r} = \tilde{\mathbf{a}}_{1,<q_r}), \mathcal{H}_{\leq q_r}) \quad (39)$$

$$P(M_{(t_p,q_{r+1})}|\operatorname{do}(\boldsymbol{A}^{\to Y}_{\leq t_p} = \tilde{\mathbf{a}}_{0,\leq t_p}, \boldsymbol{A}^{\to M}_{\leq t_p} = \tilde{\mathbf{a}}_{1,\leq t_p}), M_{[q_r,t_p]}, \mathcal{H}_{\leq q_r})$$
$$= P(M_{(t_p,q_{r+1})}|\operatorname{do}(\boldsymbol{A}^{\to Y}_{\leq t_p} = \tilde{\mathbf{a}}_{1,\leq t_p}, \boldsymbol{A}^{\to M}_{\leq t_p} = \tilde{\mathbf{a}}_{1,\leq t_p}), M_{[q_r,t_p]}, \mathcal{H}_{\leq q_r}). \quad (40)$$

Using Eqs. (35), (39) and (40), we re-write Eq. (33) as

$$P(\boldsymbol{Y}_{\mathbf{q}}[\tilde{\mathbf{a}}_0, \tilde{\mathbf{a}}_1]) = \sum_{\boldsymbol{M}_{>\tilde{t}_a}} \prod_{r=0}^{R-1} P(Y_{q_{r+1}}|\operatorname{do}(\boldsymbol{A}^{\to Y}_{<q_{r+1}} = \tilde{\mathbf{a}}_{0,<q_{r+1}}, \boldsymbol{A}^{\to M}_{<q_{r+1}} = \tilde{\mathbf{a}}_{0,<q_{r+1}}), M_{[q_r,q_{r+1})}, \mathcal{H}_{\leq q_r})$$
$$P(M_{[q_r,t_p]}|\operatorname{do}(\boldsymbol{A}^{\to Y}_{<q_r} = \tilde{\mathbf{a}}_{1,<q_r}, \boldsymbol{A}^{\to M}_{<q_r} = \tilde{\mathbf{a}}_{1,<q_r}), \mathcal{H}_{\leq q_r})$$
$$P(M_{(t_p,q_{r+1})}|\operatorname{do}(\boldsymbol{A}^{\to Y}_{\leq t_p} = \tilde{\mathbf{a}}_{1,\leq t_p}, \boldsymbol{A}^{\to M}_{\leq t_p} = \tilde{\mathbf{a}}_{1,\leq t_p}), M_{[q_r,t_p]}, \mathcal{H}_{\leq q_r}). \quad (41)$$

In Eqs. (35), (39) and (40), the target quantities are considered under 'factual' intervention pairs:

$$\operatorname{do}(\boldsymbol{A}^{\to Y}_{<\tau} = \tilde{\mathbf{a}}_{0,<\tau}, \boldsymbol{A}^{\to M}_{<\tau} = \tilde{\mathbf{a}}_{0,<\tau}) \equiv \operatorname{do}(\boldsymbol{A}_{<\tau} = \tilde{\mathbf{a}}_{0,<\tau}), \quad (42)$$
$$\operatorname{do}(\boldsymbol{A}^{\to Y}_{<\tau} = \tilde{\mathbf{a}}_{1,<\tau}, \boldsymbol{A}^{\to M}_{<\tau} = \tilde{\mathbf{a}}_{1,<\tau}) \equiv \operatorname{do}(\boldsymbol{A}_{<\tau} = \tilde{\mathbf{a}}_{1,<\tau}). \quad (43)$$

Using Eqs. (42) and (43), we can simplify Eq. (41) as follows:

$$P(\boldsymbol{Y}_{\mathbf{q}}[\tilde{\mathbf{a}}_0, \tilde{\mathbf{a}}_1]) = \sum_{\boldsymbol{M}_{>\tilde{t}_a}} \prod_{r=0}^{R-1} P(Y_{q_{r+1}}|\operatorname{do}(\tilde{\mathbf{a}}_{0,<q_{r+1}}), M_{[q_r,q_{r+1})}, \mathcal{H}_{\leq q_r})$$
$$P(M_{[q_r,t_p]}|\operatorname{do}(\tilde{\mathbf{a}}_{1,<q_r}), \mathcal{H}_{\leq q_r})$$
$$P(M_{(t_p,q_{r+1})}|\operatorname{do}(\tilde{\mathbf{a}}_{1,\leq t_p}), M_{[q_r,t_p]}, \mathcal{H}_{\leq q_r}). \quad (44)$$

In Eq. (44), both terms in the product correspond to the mediator and outcome processes under the interventions $[\tilde{\mathbf{a}}_0]$ and $[\tilde{\mathbf{a}}_1]$ respectively. These interventional distributions are identified under (**A4**), as it states that there are no unobserved confounders between the process pairs $(\boldsymbol{A}, \boldsymbol{M})$ and $(\boldsymbol{A}, \boldsymbol{Y})$. Using (**A4**) on Eq. (44), we obtain the final identification result:

$$P(\boldsymbol{Y}_{\mathbf{q}}[\tilde{\mathbf{a}}_0, \tilde{\mathbf{a}}_1]) = \sum_{\boldsymbol{M}_{>\tilde{t}_a}} \prod_{r=0}^{R-1} \underbrace{P(Y_{q_{r+1}} | \boldsymbol{A} = \tilde{\mathbf{a}}_{0,<q_{r+1}}, M_{[q_r, q_{r+1})}, \mathcal{H}_{\leq q_r})}_{\text{outcome term } \bullet}$$
$$\underbrace{P(M_{[q_r, t_p]} | \boldsymbol{A} = \tilde{\mathbf{a}}_{1,<q_r}, \mathcal{H}_{\leq q_r}) P(M_{(t_p, q_{r+1})} | \boldsymbol{A} = \tilde{\mathbf{a}}_{1,\leq t_p}, M_{[q_r, t_p]}, \mathcal{H}_{\leq q_r})}_{\text{mediator intensity } \circ}.$$

$$(45)$$

To estimate the direct and indirect effects, now the joint mediator–outcome model needs to take the treatment history $\tilde{\mathrm{a}}$ into account.

## B    Interacting Mediator–Outcome Model

Our interacting mediator–outcome model builds on marked point processes [MPP, Daley and Vere-Jones, 2003] and Gaussian processes [GP, Williams and Rasmussen, 2006]. Hence, we first introduce MPPs (B.1) and GPs (B.2). We describe our model in detail in B.3 and discuss learning and inference in B.4. Finally, we discuss computational complexity and scalability of our model in B.5.

### B.1    Marked Point Processes

A temporal point process (TPP) is a stochastic process that models continuous-time event sequences, e.g., a sequence of treatment times $\mathcal{D} = \{t_i\}_{i=1}^I$ over a period $[0, T]$ [Daley and Vere-Jones, 2003, Rasmussen, 2011]. A TPP can be represented by a counting process $N : [0, T] \to \mathbb{Z}_{\geq 0}$, which outputs the number of points until time $\tau$: $N(\tau) = \sum_{i=1}^N \mathbb{1}[t_i \leq \tau]$. We assume that there can be at most one point in an infinitesimal interval $[\tau - \Delta\tau, \tau]$: $\Delta N \in \{0, 1\}$, where $\Delta N = \lim_{\Delta\tau \downarrow 0}(N(\tau) - N(\tau - \Delta\tau))$. The counting process $N(\tau)$, and hence the change $\Delta N$, can be decomposed into a predictable compensator function $\mathbb{E}[N(\tau)] = \Lambda(\tau)$ and a zero-mean noise process (martingale) $U(\tau)$ by the Doob–Meyer theorem [Didelez, 2008]:

$$N(\tau) = \Lambda(\tau) + U(\tau) \tag{46}$$

Using the Doob–Meyer decomposition, a TPP can be uniquely determined by the expected rate of change in its counting process: $\mathbb{E}[\Delta N \mid \mathcal{H}_{<\tau}] = \Delta\Lambda \mid \mathcal{H}_{<\tau} = \lambda^*(\tau) \, \mathrm{d}\tau$, where the function $\lambda^*(\tau) = \lambda(\tau \mid \mathcal{H}_{<\tau})$ is called the conditional intensity function and the star superscript is shorthand notation for the dependence on the history $\mathcal{H}_{<\tau} = \{t_i : t_i < \tau, i = 1, \dots, I\}$.

When a temporal point carries additional information (mark, $m$), e.g., a sequence of treatment times and dosages $\mathcal{D} = \{(t_i, m_i)\}_{i=1}^I$, we model a sequence of time–mark pairs using a marked point process (MPP). Accordingly, the conditional intensity function $\lambda^*(t, m)$ is extended such that it includes the mark intensity: $\lambda^*(t, m) = \lambda^*(t)\lambda^*(m \mid t)$. Using the conditional intensity function, we can write the joint distribution of $\mathcal{D}$ as follows [Rasmussen, 2011]:

$$p(\mathcal{D}) = \prod_{i=1}^I \lambda^*(m_i \mid t_i)\lambda^*(t_i) \exp\left(-\int_0^T \lambda^*(\tau)d\tau\right). \tag{47}$$

### B.2    Gaussian Processes

A Gaussian process [GP, Williams and Rasmussen, 2006] describes a prior distribution $\mathcal{GP}$ over a continuous-valued function $f : \mathbb{R}^D \to \mathbb{R}$:

$$f(\cdot) \sim \mathcal{GP}(\mu(\cdot), k(\cdot, \cdot')), \tag{48}$$

where a mean function $\mu : \mathbb{R}^D \to \mathbb{R}$ represents the expectation $\mu(\cdot) = \mathbb{E}[f(\cdot)]$, and a symmetric positive-definite kernel function $k : \mathbb{R}^D \times \mathbb{R}^D \to \mathbb{R}$ represents the covariance $k(\cdot, \cdot') = \mathrm{cov}(\cdot, \cdot')$. In practice, we do not deal with infinite-sized distributions, rather we only consider evaluations of the

GP at a finite number of input points. The GP prior $f(\mathbf{x})$ of any subset of $N$ data points $\mathbf{x} \in \mathbb{R}^{N \times D}$ follows a joint Gaussian distribution:

$$f(\mathbf{x}) = \mathbf{f} \sim \mathcal{N}(\boldsymbol{\mu}_{\mathbf{x}}, \mathbf{K}_{\mathbf{xx}}), \tag{49}$$

where the mean $\boldsymbol{\mu}_{\mathbf{x}} = \mu(\mathbf{x}) \in \mathbb{R}^N$ is obtained by evaluating the mean function $\mu(\cdot)$ at $N$ input points and the covariance matrix $\mathbf{K}_{\mathbf{xx}} = k(\mathbf{x}, \mathbf{x}') \in \mathbb{R}^{N \times N}$ is obtained by evaluating the kernel function $k(\cdot, \cdot')$ at $N \times N$ input pairs. Commonly, we assume zero mean $\boldsymbol{\mu}_{\mathbf{x}} = \mathbf{0}$.

For a given set of $N$ training points $\mathbf{x}$ and $N_*$ test points $\mathbf{x}_*$, we can use the joint Gaussian property of GPs and write the joint distribution of $\mathbf{f} = f(\mathbf{x})$ and $\mathbf{f}_* = f(\mathbf{x}_*)$:

$$\begin{bmatrix} \mathbf{f} \\ \mathbf{f}_* \end{bmatrix} \sim \mathcal{N} \left( \mathbf{0}, \begin{bmatrix} \mathbf{K}_{\mathbf{xx}} & \mathbf{K}_{\mathbf{xx}_*} \\ \mathbf{K}_{\mathbf{x}_*\mathbf{x}} & \mathbf{K}_{\mathbf{x}_*\mathbf{x}_*} \end{bmatrix} \right), \tag{50}$$

where the cross-covariance matrix $\mathbf{K}_{\mathbf{xx}_*} \in \mathbb{R}^{N \times N_*}$ is obtained by evaluating the kernel function at $N \times N_*$ input pairs. Using the conditional distribution identities of the Gaussian distribution, we write the posterior distribution $p(\mathbf{f}_* \mid \mathbf{f})$:

$$\mathbf{f}_* \mid \mathbf{f} \sim \mathcal{N}(\mathbf{K}_{\mathbf{x}_*\mathbf{x}} \mathbf{K}_{\mathbf{xx}}^{-1} \mathbf{f}, \ \mathbf{K}_{\mathbf{x}_*\mathbf{x}_*} - \mathbf{K}_{\mathbf{x}_*\mathbf{x}} \mathbf{K}_{\mathbf{xx}}^{-1} \mathbf{K}_{\mathbf{xx}_*}). \tag{51}$$

Taking the inverse of a matrix is computationally expensive. Its computational complexity is $\mathcal{O}(N^3)$, which is unfeasible for large data sets. To make the GP inference scalable, one common method is to use inducing point approximations [Quiñonero Candela and Rasmussen, 2005]. For this, we choose a set of $M$ inducing variables $\mathbf{u} = f(\mathbf{z}) \in \mathbb{R}^M$ evaluated at $M$ inducing points $\mathbf{z} \in \mathbb{R}^{M \times D}$, where $M \ll N$. Using the same conditional Gaussian identity, we can write the conditional distribution of the function $f$ conditioned on the inducing variables $\mathbf{u}$ as follows

$$\mathbf{f} \mid \mathbf{u} \sim \mathcal{N}(\mathbf{K}_{\mathbf{xz}} \mathbf{K}_{\mathbf{zz}}^{-1} \mathbf{u}, \ \mathbf{K}_{\mathbf{xx}} - \mathbf{K}_{\mathbf{xz}} \mathbf{K}_{\mathbf{zz}}^{-1} \mathbf{K}_{\mathbf{zx}}), \tag{52}$$

where the inverse is now only required of a much smaller matrix $\mathbf{K}_{\mathbf{zz}} \in \mathbb{R}^{M \times M}$.

## B.3 Model Definition

For the joint model definition, we extend a non-parametric mediator–outcome model [Hızlı et al., 2023] to include an external intervention $A$ that jointly affects both mediators and outcomes. Similar to Hızlı et al. [2023], we combine an MPP and a conditional GP to model the interacting mediator–outcome processes. The binary value of the treatment process $A(\tau) \in \{0, 1\}$ acts as a regime indicator, which specifies whether the treatment is active or not. The structural causal model for the treatment process $\boldsymbol{A}$, the mediator process $\boldsymbol{M}$, and the outcome process $\boldsymbol{Y}$ can be written as follows:

$$A(\tau) := \mathbb{1}\{\tau \geq \tilde{t}_a\} \in \{0, 1\},$$
$$M(\tau) := f_M^{(a)}(\tau, \mathcal{H}_{<\tau}) + U_M^{(a)}(\tau), \tag{53}$$
$$Y(\tau) := f_Y^{(a)}(\tau, \mathcal{H}_{<\tau}) + U_Y^{(a)}(\tau), \tag{54}$$

where $\tau \in [0, T]$, $U_M(\tau)$ is a zero-mean noise process, and $U_Y(\tau)$ is a zero mean Gaussian noise. Eqs. (53) and (54) capture the mediator term (◉) and the outcome term (●) in Eq. (27) respectively. The mediator and outcome terms are detailed in Appendices B.3.1 and B.3.2 for a single regime $a$, since we use equivalent model definitions for both pre- and post-intervention regimes.

### B.3.1 Mediator Intensity

Considering Eq. (53) as the Doob–Meyer decomposition of the mediator process, we can interpret the function $f_M(\cdot, \cdot)$ as the compensator function $\Lambda^*(\cdot)$, and the noise $U_M$ as a zero-mean martingale. Hence, we parameterize the function $f_M(\cdot, \cdot)$ by a conditional intensity function $\lambda^*(\tau)$, where $f_M(\cdot, \cdot)$ is equivalent to the integral of $\lambda^*(\tau)$: $f_M(\tau, \mathcal{H}_{<\tau}) = \Lambda^*(\tau) = \int_{[0,T] \times \mathbb{R}} \lambda(\tau, m \mid \mathcal{H}_{<\tau}) \mathrm{d}\tau \mathrm{d}m$.

The conditional intensity function $\lambda(t_i, m_i \mid \mathcal{H}_{<t_i})$ consists of two main components: the mediator time intensity $\lambda(t_i \mid \mathcal{H}_{<t_i})$ and the mediator dosage intensity $\lambda(m_i \mid t_i, \mathcal{H}_{<t_i})$: $\lambda(t_i, m_i \mid \mathcal{H}_{<t_i}) = \lambda(t_i \mid \mathcal{H}_{<t_i})\lambda(m_i \mid t_i, \mathcal{H}_{<t_i})$. The dosage intensity is modeled as a simple GP prior with Matern-$1/2$ kernel that does not depend on the history $\mathcal{H}_{<t_i}$ and takes the absolute time as input: $\lambda(m_i \mid t_i, \mathcal{H}_{<t_i}) = \lambda(m_i \mid t_i) \sim \mathcal{GP}$.

We model the mediator time intensity $\lambda(t_i \mid \mathcal{H}_{<t_i})$ using three independent components $\{\beta_0, g_m, g_o\}$ similar to Hızlı et al. [2023], where the constant $\beta_0$ serves as a simple Poisson process baseline, the time-dependent function $g_m$ captures the dependence on the past mediators, and the time-dependent function $g_o$ captures the dependence on the past outcomes. We assume the effects of these components as additive, and their sum is squared to ensure non-negativity:

$$\lambda(\tau \mid \mathcal{H}_{<\tau}) = \big( \underbrace{\beta_0}_{\substack{\text{PP} \\ \text{baseline}}} + \underbrace{g_m(\tau; \mathbf{m})}_{\substack{\text{mediator} \\ \text{effect}}} + \underbrace{g_o(\tau; \mathbf{o})}_{\substack{\text{outcome} \\ \text{effect}}} \big)^2. \tag{55}$$

The mediator effect function $g_m(\tau; \mathbf{m})$ captures the dependence on the past mediators, e.g., the intensity should decrease just after a meal as the patient is unlikely to eat at this moment. For this, it regresses on the relative times of the last $Q_m$ mediators that occurred before time $\tau$: $(\tau - t_1, \ldots, \tau - t_{Q_m})$, where $t_q < \tau$ for $q \in 1, \ldots, Q_m$, with $g_m : \mathbb{R}_{\geq 0}^{Q_m} \to \mathbb{R}$ [Liu and Hauskrecht, 2019, Hızlı et al., 2023]. It is modeled as a GP prior $g_m \sim \mathcal{GP}$. Its input has $Q_m$ dimensions, where each dimension $q$ is modeled independently by a squared exponential (SE) kernel: $k_{g_m}(\cdot, \cdot) = \sum_{q=1}^{Q_m} k_{\text{SE}}^{(q)}(\cdot, \cdot)$.

The outcome effect function $g_m(\tau; \mathbf{m})$ captures the dependence on the past outcomes, e.g., the intensity should decrease when the blood glucose is high due to a previous meal. For this, it regresses on the relative times of the last $Q_o$ outcomes that occurred before time $\tau$: $(\tau - t_1, y_1, \ldots, \tau - t_{Q_o}, y_{Q_o})$, where $t_q < \tau$ for $q \in 1, \ldots, Q_o$, with $g_o : \{\mathbb{R}_{\geq 0} \times \mathbb{R}\}^{Q_o} \to \mathbb{R}$ [Hızlı et al., 2023]. It is modeled as a GP prior $g_o \sim \mathcal{GP}$. Its input has $Q_o \times 2$ dimensions, where $Q_o$ dimensions are modeled independently. Each dimension $q \in 1, \ldots, Q_o$ has a 2-dimensional SE kernel: $k_{g_o}(\cdot, \cdot) = \sum_{q=1}^{Q} k_{\text{SE}}^{(q)}(\cdot, \cdot)$: the first axis represents the relative-time input and the second axis represents the measurement value, e.g., glucose levels. For more details on the non-parametric point process model, see Hızlı et al. [2023].

### B.3.2 Outcome Model

For Eq. (54), we model the function $f_Y(\cdot, \cdot)$ using two components: (i) a baseline function $f_b(\tau)$ that is independent of the history (past mediators) and (ii) a mediator response function $f_m(\cdot, \cdot)$ that captures the outcome response due to a mediator occurrence [Schulam and Saria, 2017, Cheng et al., 2020, Hızlı et al., 2023]. Together with the independent, zero-mean Gaussian noise $u_Y \sim \mathcal{N}(0, \sigma_Y^2)$, we can write Eq. (54) as follows:

$$Y(\tau) = \underbrace{f_b(\tau)}_{\text{baseline}} + \underbrace{f_m(\tau; \mathbf{m})}_{\text{mediator response}} + \underbrace{u_Y(\tau)}_{\text{noise}}. \tag{56}$$

The baseline function has a GP prior $f_b \sim \mathcal{GP}$. Its kernel is a sum of a constant and a periodic (PER) kernel $k_b(\cdot, \cdot) = k_{\text{CONST}}(\cdot, \cdot) + k_{\text{PER}}(\cdot, \cdot)$. The mediator response function $f_m$ models the dependence of the future outcomes on the past mediators. We assume that the effect of the past mediators is additive, and the magnitude and the shape of the response are factorized as follows:

$$f_m(\tau; \mathbf{m}) = \sum_{i:t_i \leq \tau} l(m_i) f_m^0(\tau - t_i), \tag{57}$$

where (i) the magnitude of the response depends on the mark $m_i$ through the linear function $l(m_i)$, and (ii) the shape of the response depends on the relative time $\tau - t_i$ through a GP prior $f_m^0 \sim \mathcal{GP}$ that has a SE kernel: $k_m(\cdot, \cdot) = k_{\text{SE}}(\cdot, \cdot)$. We ensure that a future mediator does not affect a past outcome by using a 'time-marked' kernel [Cunningham et al., 2012] where the output of the kernel is set to zero if the relative time input is negative, i.e., the mediator has occurred after the outcome, similar to Cheng et al. [2020], Hızlı et al. [2023]. Besides, we assume that the mediator has a causal effect on the outcome for an effective period $\mathcal{T}_{\text{eff}}$, e.g., the glucose response of a meal takes place in the next $3\,\text{h}$ [Wyatt et al., 2021]. For more details on the non-parametric outcome model, see Hızlı et al. [2023].

### B.3.3 Joint Distribution of Mediator–Outcome Model

In our problem setup, each patient undergoes the surgery and is observed in two regimes (pre- and post-surgery): $\mathcal{D} = \{\mathcal{D}^{(a)}\}_{a \in \{0,1\}}$. Within each regime $a \in \{0, 1\}$, the data set $\mathcal{D}^{(a)}$ contains the measurements of mediator $M$ and outcome $Y$ at irregular times: $\{(t_i^{(a)}, m_i^{(a)})\}_{i=1}^{I^{(a)}}$ and

$\{(t_j^{(a)}, y_j^{(a)})\}_{j=1}^{J^{(a)}}$. For completeness, we repeat the joint distribution of $\mathcal{D}$:

$$p(\mathcal{D}) = \prod_{a \in \{0,1\}} \left[ \exp(-\Lambda^{(a)}) \prod_{i=1}^{I^{(a)}} \underbrace{\lambda^{(a)}(t_i, m_i \mid \mathcal{H}_{<t_i})}_{\text{mediator intensity } \circ} \prod_{j=1}^{J^{(a)}} \underbrace{p^{(a)}(y_j \mid t_j, \mathcal{H}_{<t_j})}_{\text{outcome model } \bullet} \right], \quad (58)$$

### B.4 Learning and Inference

We use the likelihood in Eq. (58) to learn the joint mediator–outcome model. The mediator intensity has a sparse GP prior using inducing point approximations [Hensman et al., 2015]. Hence, we use variational inference and optimize the evidence lower bound (ELBO). The outcome model is a standard GP prior. Hence, we use the marginal likelihood to learn the hyperparameters. For more details on the derivations of the learning algorithms, see Hızlı et al. [2023].

Following Eq. (58), we learn a single joint model for each of pre-surgery ($a = 0, [\varnothing, \varnothing]$) and post-surgery regimes ($a = 1, [\tilde{t}_a, \tilde{t}_a]$). With these two models, we can estimate the two interventional trajectories $\mathbf{Y}_{>\tilde{t}_a}[\varnothing, \varnothing]$ and $\mathbf{Y}_{>\tilde{t}_a}[\tilde{t}_a, \tilde{t}_a]$ under 'factual' interventions. For the interventional trajectory $\mathbf{Y}_{>\tilde{t}_a}[\varnothing, \tilde{t}_a]$ under the hypothetical intervention $[\varnothing, \tilde{t}_a]$, Eq. (27) states that the outcomes follow the pre-surgery outcome distribution while the mediators follow the post-surgery distribution. Therefore, we combine the pre-surgery outcome model $p(\mathbf{Y}[A = \varnothing])$ and the post-surgery mediator intensity $\lambda^*[A = \tilde{t}_a]$ to estimate the interventional trajectory $\mathbf{Y}_{>\tilde{t}_a}[\varnothing, \tilde{t}_a]$.

### B.5 Scalability

**Outcome Model.** The computational complexity for the learning and inference in standard GPs is $\mathcal{O}(N^3)$ where $N$ is the number of training data points. In the RCT data set used in experiments, we have ca. 5100 data points for each period, resulting from 17 patients with 3-day long glucose measurements in approximately 15-minute intervals. This is a relatively large data set that renders the standard GP inference slow. However, a straightforward inducing point approximation would not work on the outcome model, due to the 'time-marked' mediator-response function where a past mediator only affects the future outcomes. To make this approach more scalable, an efficient inducing point implementation will be required and it will be considered as future work. We believe that this will be a meaningful contribution for scalable Bayesian estimation of treatment-response curves [Schulam and Saria, 2017, Xu et al., 2016, Soleimani et al., 2017, Cheng et al., 2020].

**Mediator Intensity.** The computational complexity for the learning and inference in GPs with inducing point approximations is $\mathcal{O}(NM^2)$ where $M$ is the number of inducing points. In our experiments, we see that $M$ can be chosen in the order of 10 and we use $M = 20$ in all experiments. Hence, the learning and the inference for the mediator intensity is more scalable compared to the outcome model.

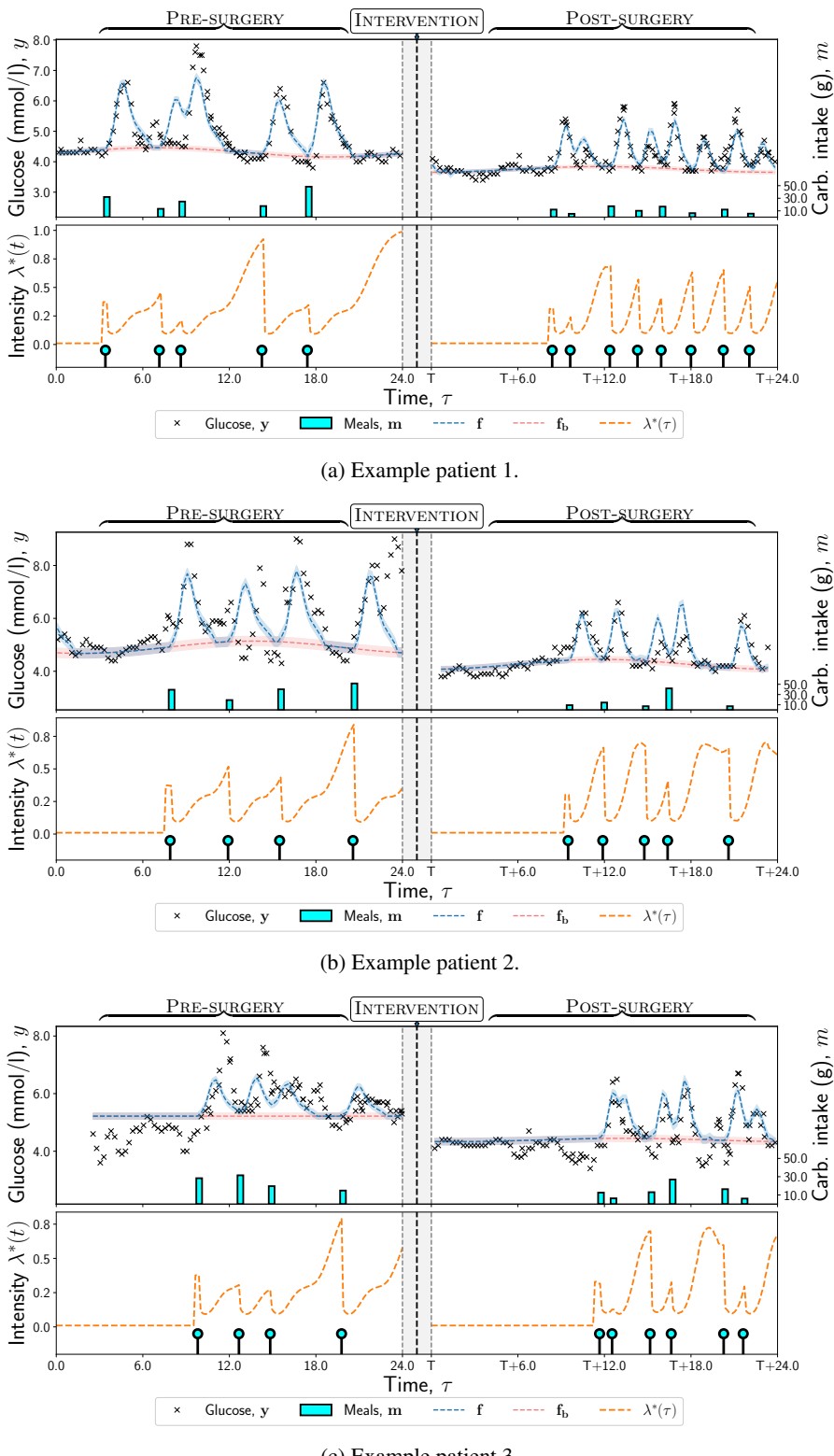

(a) Example patient 1.

(b) Example patient 2.

(c) Example patient 3.

Figure 7: Example 1-day long meal-glucose data and joint model fits for three patients: meals (carbohydrate intake, cyan bars), glucose (black crosses), predicted meal intensity $\lambda^*$ (orange dashed line), predicted glucose baseline $\mathbf{f_b}$ (red dashed line) and predicted glucose progression $\mathbf{f}$ (blue dashed line). For blood glucose, we see, *after the surgery*, (i) the baseline declines, and (ii) a meal produces a faster rise and decline. For meals, we see, *after the surgery*, the patient eats (i) more frequently and (ii) less carbohydrate per meal.

# C   Experiment Details

## C.1   Real-world Study

For the real-world study, we consider data from a real-world randomized controlled trial (RCT) about the effects of bariatric surgery on blood glucose [Saarinen et al., 2019, Ashrafi et al., 2021]. We investigate whether our model can learn clinically-meaningful direct and indirect effects of bariatric surgery on blood glucose, by analyzing how surgery affects glucose through the changed diet (indirectly) or other metabolic processes (directly).

**Dataset.**   The RCT dataset consists of surgery–meal–blood glucose (*treatment–mediator–outcome*) measurements of 15 obesity patients (body mass index, BMI $\geq 35\,\mathrm{kg/m^2}$) undergoing a gastric bypass surgery. In the original study, patients are randomized into two types of surgeries: Roux-en-Y gastric bypass (RYGB) [Wittgrove et al., 1994] and one-anastomosis gastric bypass (OAGB) [Rutledge, 2001]. Nevertheless, we consider both as a single surgery type since it has been previously reported that two surgeries do not have a statistically-significant difference between their effects on the blood glucose progression [Ashrafi et al., 2021]. A continuous-monitoring device measures the blood glucose of each patient at 15-minute intervals. Patients record their meals in a food diary. Data is collected over two 3-day long periods: (i) the *pre-surgery* period, which is 2-months prior to the surgery at weight stability, and (ii) the *post-surgery* period, which takes place right after the surgery. From the surgery-meal-glucose data set, we show 1-day long meal-glucose measurements in pre- and post-surgery periods for three example patients in Fig. 7.

**Data Preprocessing.**   Since patients record their meals with error, self-recorded food diaries contain very noisy observations, especially in terms of the meal times. For example, a patient can record a meal with a 30-min delay when the blood glucose already peaked to its local maxima (top of the meal bump). Or, they can record a single meal as multiple meal records. This may cause a naive supervised learning algorithm to come to a counterintuitive conclusion that a meal causes a temporary decrease in the blood glucose. Therefore, we perform a data preprocessing step on the meals to correct their occurrence times. To avoid multiple records, we combine multiple meals that occur within $30\,\mathrm{min}$. To align the meal response bumps and the meal times, i.e., to find the true meal time values, we use a Bayesian error-in-variables model that accounts for the errors in meal timings [Ashrafi et al., 2021, Zhang et al., 2021]. We train the error-in-variables model using the provided code in Stan [Carpenter et al., 2017] and use the posterior means of the true meal times.

**Model.**   We model the pre- and post-surgery sequences of meals and blood glucose measurements using our interacting mediator–outcome model, given in Eqs. (55) and (56). We represent each of pre- and post-surgery periods as a single regime $a \in \{0, 1\}$ using the binary-valued treatment process $A(\tau) \in \{0, 1\}$. In the following, we explain the model parameters for a single regime $a \in \{0, 1\}$, since we use the same model definition and initialisation for both regimes.

The mediator dosage intensity $\lambda(m \mid \tau) \sim \mathcal{GP}$ has a Matern-$1/2$ kernel with the variance and the lengthscale parameters are initialized as follows: $\sigma_d^2 = 1.0, \ell_d = 1.0$. Its hyperparameters are learned. The mediator time intensity has three components: $\{\beta_0, g_m, g_o\}$. The hyperparameters of these components are set using the domain knowledge. The Poisson process baseline $\beta_0$ is initialized to $0.1$. The meal-effect function $g_m$ depends on the relative time to the last meal: $Q_m = 1$. The variance and the lengthscale parameters of its SE kernel are initialized as: $\sigma_m^2 = 0.1, \ell_m = 1.5$. The lengthscale value $1.5$ captures the likely decline and rise in the meal intensity right after a previous meal in the next $5\,\mathrm{h}$. Similarly, the glucose-effect function $g_o$ depends on the relative time and value of the last glucose measurement: $Q_o = 1$. The hyperparameters of its 2-dimensional SE kernel are initialized as: $\sigma_o^2 = 0.1, \ell_o = [100.0, 5.0]$. The large lengthscale value $100.0$ in the time dimension encourages a piecewise linear function that changes with a new glucose measurement in every $\sim15\,\mathrm{min}$, since the glucose measurements are more frequent compared to the meal measurements ($\sim3\,\mathrm{h}$). The large lengthscale value $5.0$ of the glucose dimension encourages a smooth, slow-changing function as blood glucose levels fluctuate in the range $[4.0, 9.0]$ to capture a monotonically decreasing intensity as the blood glucose levels increase. The number of inducing points are chosen as $M = 20$ and they are placed on a regular grid in the input space. The hyperparameters are not learned, while the inducing variables $\mathbf{Z}$ are learned. The mediator intensity is shared among all patients.

The outcome model has three components: $\{f_b, f_m, u_Y\}$. The most important hyperparameters are the GP lengthscale parameters, which are selected by a combination of domain knowledge and validation of the model fit. For meal-response, we considered lengthscales $\{0.15\,\text{h}, 0.3\,\text{h}, 0.5\,\text{h}, 1.0\,\text{h}\}$, and for the baseline, lengthscales $\{5\,\text{h}, 10\,\text{h}, 20\,\text{h}\}$.

The kernel of the baseline function $f_b$ is equal to the sum of a constant and a periodic kernel. The intercept parameter of the constant kernel is initialized to $1.0$. The period parameter $p$ is set to $24\,\text{h}$ so that $f_b$ models the daily glucose profile. The variance and the lengthscale parameters of the $k_b$ are initialized as: $\sigma_b^2 = 1.0, \ell_b = 10.0$. The large lengthscale value $10.0$ encourages a slow-changing, smooth baseline function. We assume that each patient has an independent baseline function $f_b$. The meal response function $f_m = \sum_{i:t_i \leq \tau} l(m_i) f_m^0(\tau - t_i)$ consists of two functions: (i) a linear function $l(\cdot)$ whose intercept and slope parameters are initialized to $0.1$, and (ii) the response shape function $f_m^0 \sim \mathcal{GP}$ with a SE kernel whose variance and lengthscale parameters are initialized as: $\sigma_0^2 = 1.0, \ell_0 = 0.5$. The magnitude function $l(m_i)$ is modeled in a hierarchical manner, where each patient has their own intercept and slope parameters and they all share hierarchical Gaussian priors $N(0, 0.1^2)$. The shape function $f_m^0$ is shared among the patients as the meal responses for different individuals are very similar. Furthermore, the effective period for a meal $\mathcal{T}_{\text{eff}}$ is set to $3\,\text{h}$ following the empirical findings on the duration of the meal response in Wyatt et al. [2021]. The hyperparameters of the outcome model are learned except the period of the baseline $p$ ($24\,\text{h}$) and the effective period $\mathcal{T}_{\text{eff}}$ of the shape function ($3\,\text{h}$), which are chosen w.r.t. the domain knowledge.

**Training.** As described above, the interacting mediator–outcome model is trained on the surgery-meal-glucose dataset. For each regime, one set of joint model is learned. We train the hyperparameters of the meal dosage intensity using the marginal likelihood and the inducing variables of the meal time intensity using the ELBO. For the outcome model, we learn the hyperparameters using the marginal likelihood. We show three example training fits of the joint model on 1-day long meal-glucose measurements in pre- and post-surgery periods in Fig. 7.

**Next Meal Predictions in Section 5.1.3.** Once the models are learned, we can sample from the GP posteriors. We take 5000 samples from the posterior of the time intensity for the next meal time under a typical glucose level. Similarly, we take 5000 samples from the posterior of the dosage intensity for the carbohydrate intake per meal. In Fig. 5 of the main text, we show the kernel density estimations (KDE) using dashed lines. For the computation of the KDE, we use the built-in function of the 'seaborn' library.

**Direct and Indirect Effects on Glycemia in Section 5.1.4.** In Section 5.1.4, we investigate the direct and indirect effects of the surgery on the glycemia to answer the following causal query: *how much of the surgery effect can be contributed to the changed diet*? To measure the contribution of the direct and indirect effects, we use two metrics: (i) the percentage time spent in hypoglycemia (HG) $\%T_{\text{HG}} = \{t : Y(t) \leq 3.9\,\text{mmol/l}\}/T$, and (ii) the percentage time spent in above-normal-glycemia (ANG) $\%T_{\text{ANG}} = \{t : Y(t) \geq 5.6\,\text{mmol/l}\}/T$. In the following, we describe the computation for one metric for brevity, e.g., $\%T_{\text{HG}}$, as the computations of the two metrics are the same.

The metrics can be calculated on the surgery-meal-glucose data set in a straightforward manner. After we calculate them for the pre-surgery period $[t_A = \varnothing] \equiv [\varnothing, \varnothing]$ and the post-surgery period $[t_A = \tilde{t}_a] \equiv [\tilde{t}_a, \tilde{t}_a]$, we compute the total causal effect (TE) as follows

$$\text{TE}_{\text{HG}}(\tilde{t}_a) = \%T_{\text{HG}}[\tilde{t}_a, \tilde{t}_a] - \%T_{\text{HG}}[\varnothing, \varnothing].$$

For direct and indirect effects, we further need to estimate $\%T_{\text{HG}}[\varnothing, \tilde{t}_a]$, i.e., the percentage time spent in hypoglycemia under a hypothetical intervention $[\varnothing, \tilde{t}_a]$. As the counterfactual $\%T_{\text{HG}}[\varnothing, \tilde{t}_a]$ is not available in closed form, we take samples from the posterior of the learned joint model and perform a Monte Carlo approximation. As the posterior samples, we sample 3-day long meal-glucose trajectories for 17 patients for each of the three interventional regimes: $\{[\varnothing, \varnothing], [\varnothing, \tilde{t}_a], [\tilde{t}_a, \tilde{t}_a]\}$. Each 1-day long trajectory has 40 glucose measurements, which makes 2040 samples for each regime. For the interventional trajectories to be comparable, we fix the noise variables of the point process sampling, similar to Hızlı et al. [2023]. Then, the Monte Carlo approximations of the NDE and the

Table 3: Benchmark models considered in the semi-synthetic study.

| JOINT MODEL | MODEL COMPONENTS | | |
|---|---|---|---|
| | $A \rightarrow Y$ | MEDIATOR | RESPONSE |
| M1 | ✓ | Non-interacting (L15) | Parametric (S17) |
| M2 | ✓ | Non-interacting (L15) | Non-parametric (H22) |
| M3 | ✓ | Interacting (H22) | Parametric (S17) |
| Z21-1 | ✓ | Interacting (H22) | Parametric (Z21) |
| Z21-2 | ✓ | ORACLE | Parametric (Z21) |
| H22 | ✗ | Interacting (H22) | Non-parametric (H22) |
| OUR | ✓ | Interacting (H22) | Non-parametric (H22) |

NIE in $\%T_{\mathrm{HG}}$ are computed as follows

$$\mathrm{NDE}_{\mathrm{HG}}(\tilde{t}_a) = \sum_{i=1}^{17 \times 3} \%T_{\mathrm{HG}}^{(i)}[\tilde{t}_a, \tilde{t}_a] - \%T_{\mathrm{HG}}^{(i)}[\varnothing, \tilde{t}_a],$$

$$\mathrm{NIE}_{\mathrm{HG}}(\tilde{t}_a) = \sum_{i=1}^{17 \times 3} \%T_{\mathrm{HG}}^{(i)}[\varnothing, \tilde{t}_a] - \%T_{\mathrm{HG}}^{(i)}[\varnothing, \varnothing],$$

where each index $i \in 1, \ldots, 17 \times 3$ refers to a 1-day long trajectory sample.

## C.2  Semi-synthetic Study

The performance on the causal tasks are validated on synthetic studies, since the true causal effects are unknown in real-world observational data sets. Hence, we set up a realistic, semi-synthetic simulation study to evaluate the performance of our model on two causal tasks: estimating the NDE and NIE trajectories.

### C.2.1  Simulator

To set up a realistic semi-synthetic study, we design realistic ground-truth simulators, that are based on the real-world data set of the RCT study. To achieve this, we train our joint mediator–outcome model on the surgery-meal-blood glucose data to obtain a single joint simulator for each regime $a \in \{0, 1\}$. We use the same model definition, initialisation and learning procedure in Appendix C.1. For a visual example on how the data and the simulator fits look, see Figs. 7 and 8.

The ground-truth joint simulator has two components: the meal (mediator) simulator and the glucose (outcome) simulator. To train the meal simulator, we use the meal-glucose data from all patients. We show the training fits on 1-day long meal trajectories of three example patients in the bottom panels of the sub-figures in Fig. 7. In addition, we show how the function components $g_m$ and $g_o$ differs in Figs. 9a and 9b. We see that in both pre- and post-surgery periods, the meal-time intensity decreases after a meal event. Besides, the post-surgery meal-time intensity increases faster compared to the pre-surgery meal-time intensity as expected, as the surgery leads to a diet with more frequent, smaller meals. For the decrease in the size of the meals, we show how the carbohydrate intake (meal dosage) intensity changes after the surgery in Fig. 9c.

For the glucose simulator, we use the meal-glucose data for three patients. This enables individualization between patients through their glucose baseline $f_b(\cdot)$ and the meal response magnitude function $l(\cdot)$. We chose to train the glucose simulator on three patients instead of all since (i) the usage of all patients make the posterior sampling slow due to the increase in the simulator size and (ii) three patients are enough to create a realistic individualization behavior where we have three patient groups with similar glycemia: $(\mathrm{gr}_1, \mathrm{gr}_2, \mathrm{gr}_3)$. We show 1-day long examples of the outcome simulator fit in Fig. 8. Besides, we show how the glucose baseline $f_b$ and the meal response $f_m$ differs between these three patients in Fig. 10.

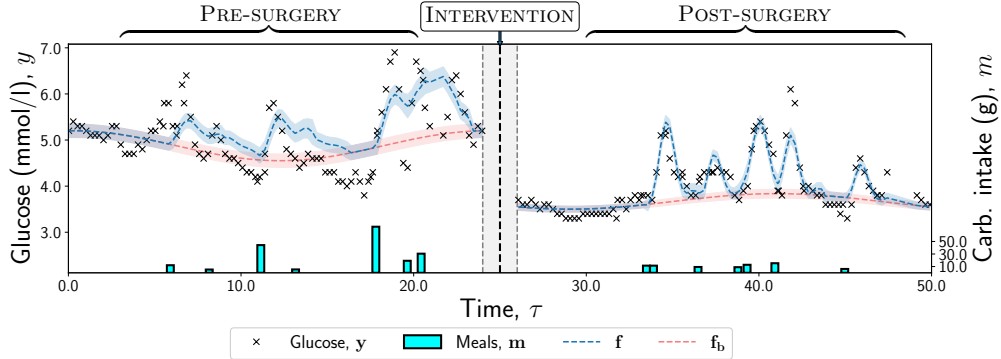

(a) Example 1-day long glucose predictions of patient group 1 for the outcome simulator.

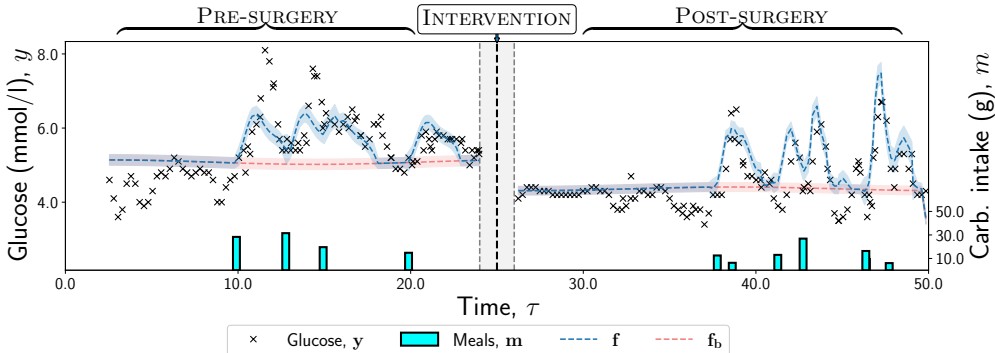

(b) Example 1-day long glucose predictions of patient group 2 for the outcome simulator.

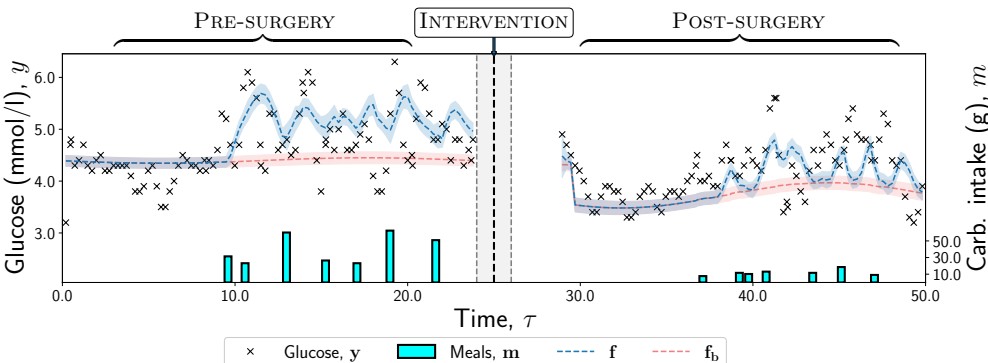

(c) Example 1-day long glucose predictions of patient group 3 for the outcome simulator.

Figure 8: Example 1-day long meal-glucose data and outcome simulator fits for three patient patient groups: meals (carbohydrate intake, cyan bars), glucose (black crosses), predicted glucose baseline $\mathbf{f_b}$ (red dashed line) and predicted glucose progression $\mathbf{f}$ (blue dashed line).

### C.2.2 Benchmarks

In the semi-synthetic study, we use the benchmark models shown in Table 3. In the following, we describe the model definitions and implementation details of the mediator and outcome models.

**Non-interacting mediator model (L15).** The non-interacting mediator model [Non-interacting (L15), Lloyd et al., 2015] is a GP-modulated non-homogeneous Poisson process where the intensity $\lambda(\tau) = f^2(\tau)$ is the square transformation of a latent intensity $f \sim \mathcal{GP}$ with a GP prior. The GP model uses inducing point approximations for scalable inference and variational inference is used for learning. We use an implementation of the model in GPflow [Matthews et al., 2017].

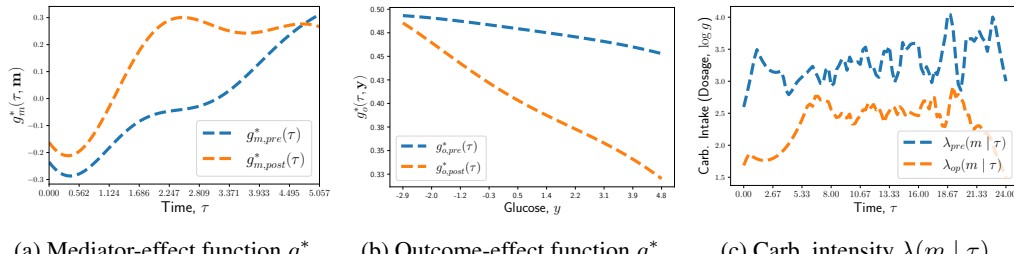

(a) Mediator-effect function $g_m^*$.  (b) Outcome-effect function $g_o^*$.  (c) Carb. intensity $\lambda(m \mid \tau)$.

Figure 9: Comparison of the ground-truth meal simulator intensity components between pre- and post-surgery periods: **(a)** the mediator-effect function $g_m^*$ that models the dependence of a future meal on past meals, **(b)** the outcome-effect function $g_o^*$ that models the dependence of a future meal on past glucose levels, and **(c)** the carbohydrate intake intensity $\lambda(m \mid \tau)$.

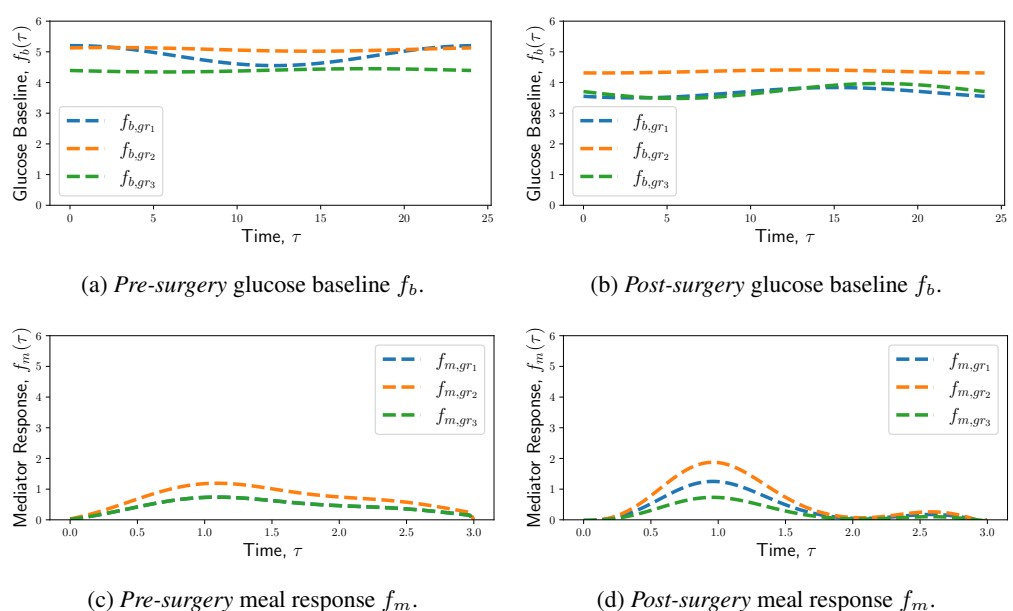

(a) *Pre-surgery* glucose baseline $f_b$.  (b) *Post-surgery* glucose baseline $f_b$.

(c) *Pre-surgery* meal response $f_m$.  (d) *Post-surgery* meal response $f_m$.

Figure 10: Comparison of the ground-truth glucose simulator components between pre- and post-surgery periods for three patient groups $\{gr_1, gr_2, gr_3\}$ corresponding to three real-world patients. For three groups, we show **(a)** *pre-surgery* glucose baseline $f_b$, **(b)** *post-surgery* glucose baseline $f_b$, **(c)** *pre-surgery* meal response function $f_m$, and **(d)** *post-surgery* meal response function $f_m$.

**Parametric mediator-response (S17).** As a simple benchmark for the outcome model, we add a highly-cited parametric response model [Parametric (S17), Schulam and Saria, 2017]. It has a conditional GP prior with a parametric response function that models the mediator response in the outcome as a constant effect. We use the implementation provided in the supplementary material in Schulam and Saria [2017].

**Parametric mediator-response (Z21).** Zeng et al. [2021] proposed a longitudinal causal mediation model based on functional principal component analysis, where its response model [Parametric (Z21), Zeng et al., 2021] is a simple linear function with a single slope parameter. The principal components are chosen as a linear combination of a spline basis where the coefficients satisfy the orthogonality constraint. They assume that the treatment intervention only affects the coefficients of the spline basis. In their work, Zeng et al. [2021] consider the mediator as a continuous-valued stochastic process that is observed at the same times with the outcome measurements, which is different than our problem setup where the mediator is a marked point process. Therefore, we combine the outcome model of [Zeng et al., 2021] with a point-process mediator model in the joint benchmark models Z21-1 and Z21-2, and transform the meal sequence into a continuous-valued sequence that is observed at the same times with the outcome measurements. For this, we use the carbohydrate intake values of the

meals in their effective periods and we simply add up the carbohydrate values if two meals are nearby similar to the additive mediator response assumption in our outcome model definition. We use the publicly available code implemented in R. For inference, the method uses Markov chain Monte Carlo (MCMC). We use 20000 samples with a burn-in period of 2000 samples.

**Non-parametric model (H22).** The non-parametric model is a GP-based joint model that combines an interacting mediator model [Interacting (H22), Hızlı et al., 2023] and a non-parametric response model [Non-parametric (H22), Hızlı et al., 2023]. The model assumes that the intervention effect is fully-mediated through meals, and hence it can not capture the direct arrows from the treatment to the outcomes ($A\text{--}\rightarrow Y$). Without explicitly modeling these direct effects, the learned outcome model seems to average the baseline and response functions of the pre- and post-surgery periods. We use an implementation of the model in GPflow [Matthews et al., 2017].

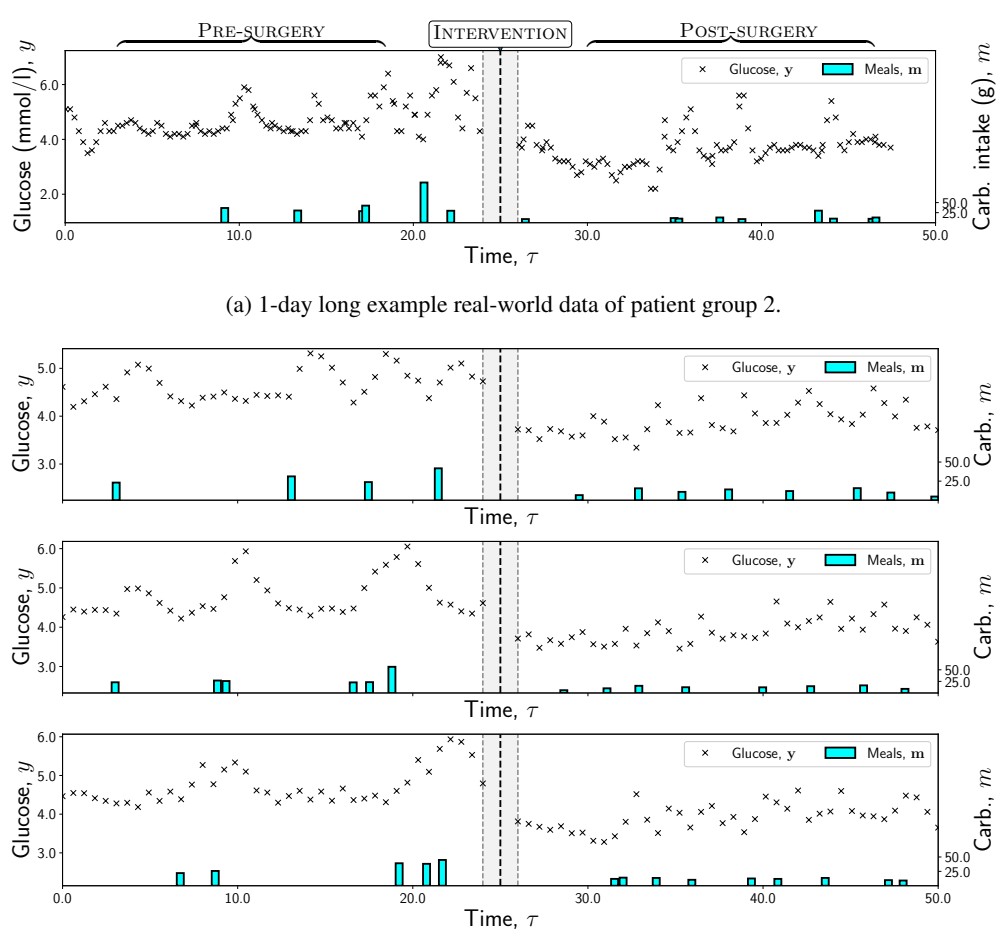

(a) 1-day long example real-world data of patient group 2.

(b) 1-day long example semi-synthetic data for patient group 2.

Figure 11: Example semi-synthetic trajectories from the training set. **(a)** Example 1-day long real-world data for patient group 2. **(b)** Example 1-day long semi-synthetic data for patient group 2.

### C.2.3  Semi-Synthetic Train and Test Trajectories

Using the ground-truth meal-glucose simulator, we sample 1-day long meal-glucose trajectories of $50$ synthetic patients for both pre- and post-surgery periods, i.e., for each regime $a \in \{0, 1\}$. We show some examples of the simulated trajectories in Fig. 11. We train the benchmark models on both periods, and hence obtain a single joint model for each regime $a \in \{0, 1\}$.

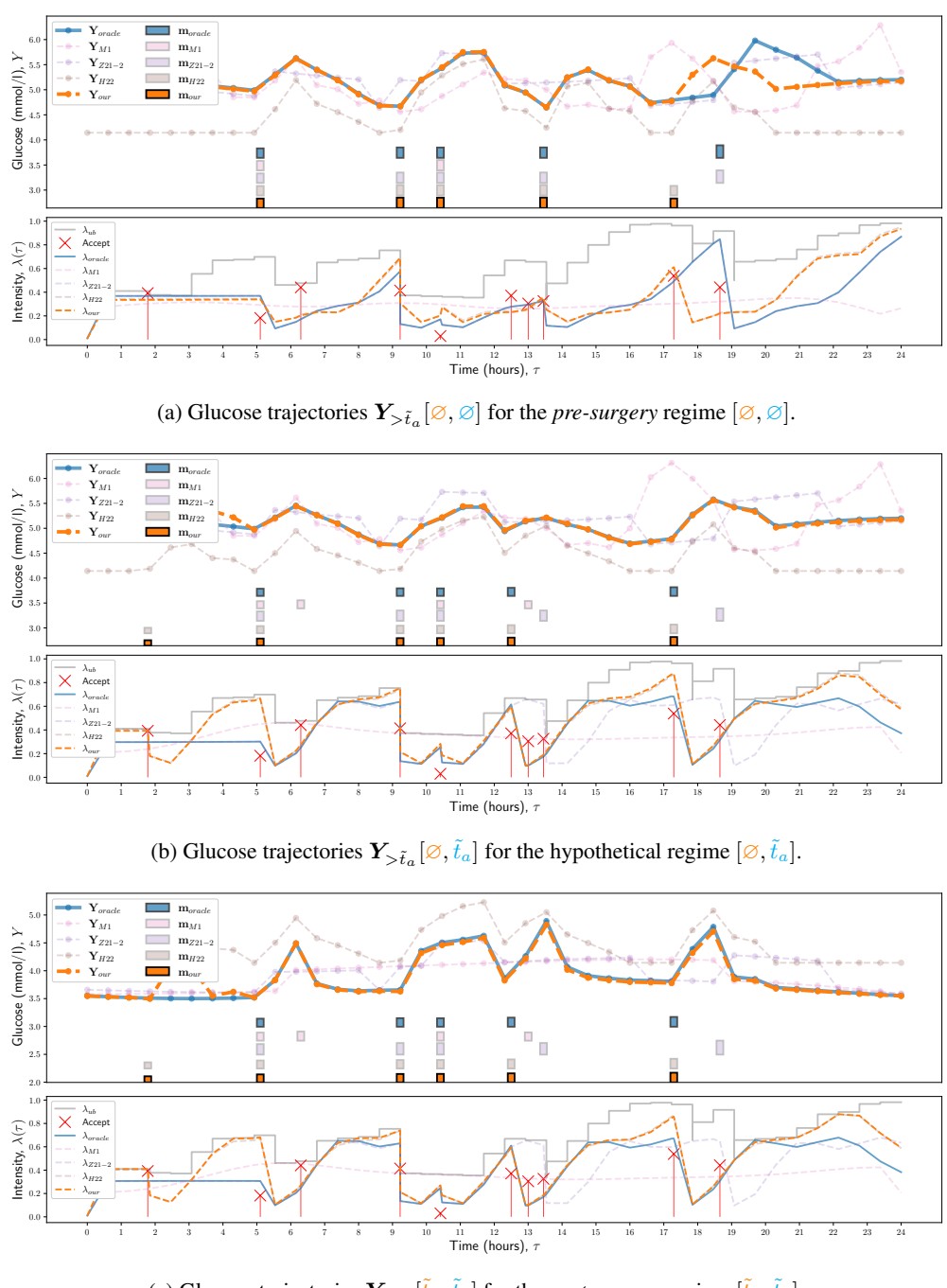

(a) Glucose trajectories $\boldsymbol{Y}_{>\tilde{t}_a}[\varnothing, \varnothing]$ for the *pre-surgery* regime $[\varnothing, \varnothing]$.

(b) Glucose trajectories $\boldsymbol{Y}_{>\tilde{t}_a}[\varnothing, \tilde{t}_a]$ for the hypothetical regime $[\varnothing, \tilde{t}_a]$.

(c) Glucose trajectories $\boldsymbol{Y}_{>\tilde{t}_a}[\tilde{t}_a, \tilde{t}_a]$ for the *post-surgery* regime $[\tilde{t}_a, \tilde{t}_a]$.

Figure 12: Example 1-day long interventional (test) meal-glucose trajectories of the oracle model, the proposed model and the benchmark models in the semi-synthetic study for three hypothetical interventional regimes $[\varnothing, \varnothing]$, $[\varnothing, \tilde{t}_a]$ and $[\tilde{t}_a, \tilde{t}_a]$. In all figures, (i) at top panels we show the meal-glucose trajectories and (ii) at bottom panels we show the meal time intensities and the point process sampling procedure. **(a)** The *pre-surgery* glucose trajectories $\boldsymbol{Y}_{>\tilde{t}_a}[\varnothing, \varnothing]$ for the *pre-surgery* regime $[\varnothing, \varnothing]$. **(b)** The hypothetical glucose trajectories $\boldsymbol{Y}_{>\tilde{t}_a}[\varnothing, \tilde{t}_a]$ for the hypothetical regime $[\varnothing, \tilde{t}_a]$. **(c)** The *post-surgery* glucose trajectories $\boldsymbol{Y}_{>\tilde{t}_a}[\tilde{t}_a, \tilde{t}_a]$ for the *post-surgery* regime $[\tilde{t}_a, \tilde{t}_a]$.

Once the models are trained, we sample three 1-day long interventional trajectories $\boldsymbol{Y}_{>\tilde{t}_a}[\varnothing, \varnothing]$, $\boldsymbol{Y}_{>\tilde{t}_a}[\varnothing, \tilde{t}_a]$ and $\boldsymbol{Y}_{>\tilde{t}_a}[\tilde{t}_a, \tilde{t}_a]$, corresponding to three hypothetical interventional regimes $[\varnothing, \varnothing]$,

$[\varnothing, \tilde{t}_a]$ and $[\tilde{t}_a, \tilde{t}_a]$. Three trajectories are sampled for the benchmarks and our model. These estimated trajectories are compared to the ground-truth trajectories sampled from the ground-truth simulator. To sample meal-glucose trajectories that are comparable w.r.t. their meal times, we fix the noise variables of the point process sampling similar to Hızlı et al. [2023]. After the interventional trajectories are sampled, we compute the $\text{TE}(\tilde{t}_a)$, $\text{NDE}(\tilde{t}_a)$ and $\text{NIE}(\tilde{t}_a)$ as described in Eqs. (15) to (17). We show a 1-day long example for the three trajectories in Fig. 12. For all three trajectories $\boldsymbol{Y}_{>\tilde{t}_a}[\varnothing, \varnothing]$, $\boldsymbol{Y}_{>\tilde{t}_a}[\varnothing, \tilde{t}_a]$ and $\boldsymbol{Y}_{>\tilde{t}_a}[\tilde{t}_a, \tilde{t}_a]$, we see that our model follows the oracle meal intensity and the glucose trajectory well. Hence, our model also follows the direct, indirect and total effect trajectories well, since these effects are calculated by subtracting the three interventional trajectories.

We can inspect Fig. 12 to understand how the considered benchmark models fit to the synthetic training data. We see that the simple intercept response model of **S17** does not capture the meal response curves well, e.g., the constant response (pink dashed line) in Fig. 12c roughly between 5am - 6pm (in the effective meal periods). Similarly, the linear mediator response function of **Z21** is unable to capture the complex, non-linear meal response, e.g., simple constant responses in the purple dashed line in Fig. 12c for the time periods $[5, 8]$, $[9, 17]$ and $[19, 22]$. We see that the non-parametric benchmark model **H22** has a glucose baseline and a meal response that is averaged over the pre- and post-surgery periods, e.g., the under-oracle glucose baseline of **H22** in Fig. 12a and the above-oracle glucose baseline of **H22** in Fig. 12c (brown dashed lines). Similarly, we see the model fit for the non-interacting mediator benchmark model (pink dashed intensity lines in bottom panels of Figs. 12a to 12c), and it does not capture the dynamic, long-range dependence between the past meal-glucose measurements and the future meals.

# D  Additional Results

## D.1  Sensitivity Analysis

In this section, we perform sensitivity analysis by creating a hypothetical hidden-confounding scenario in which the causal assumptions do no longer hold. In such a scenario, a hidden confounder that affects both the mediator and the outcome sequences also can be seen as a source of model misspecification. Therefore, our goals are to understand: (i) how sensitive the model predictions are to the causal assumptions, and (ii) how our model behaves under model misspecification.

**Setup.**  To create a hypothetical hidden-confounding scenario, we assume that patients are advised to be more active after the surgery in the observational (training) data. We sample an unobserved binary confounder $C_i$ for being active during the period of each meal $i$ with probability $\beta$: $C_i \sim \mathcal{BE}(\beta)$. If active in the meal period, a patient both (i) eats more (bias in meal dosage $m$: $m_{\text{active}} = m^2/2$), and (ii) their glucose peaks less (bias in response to a meal via the response magnitude $m_{\text{effective}} = m^2/4$). We vary the amount of confounding by considering $\beta \in \{0.0, 0.2, 0.4, 0.6, 0.8, 1.0\}$, where $\beta = 0$ corresponds to the case of no unobserved confounding (NUC). Under this scenario, the models will learn biased estimates for meal amounts and meal responses.

We consider three different interventional distributions at test time to understand how the models behave under unbiased or biased estimates: (i) same as the observational data where the patients are advised to be active with the same probability $\beta$ in the *post-surgery* period, (ii) the patients are advised to be active with the same probability $\beta$ in the *pre-surgery* period, and (iii) the patients are not advised at all. The interventional distribution in (i) is the closest to the observational distribution, while the mismatch between the observational distribution and the interventional distributions in (ii) and (iii) increase for larger $\beta$ values.

**Results.**  For each case, we report the MSE in the natural direct effect $\text{NDE}(\tilde{t}_a) = \boldsymbol{Y}_{>\tilde{t}_a}[\tilde{t}_a, \tilde{t}_a] - \boldsymbol{Y}_{>\tilde{t}_a}[\varnothing, \tilde{t}_a]$, the natural indirect effect $\text{NIE}(\tilde{t}_a) = \boldsymbol{Y}_{>\tilde{t}_a}[\varnothing, \tilde{t}_a] - \boldsymbol{Y}_{\tilde{t}_a}[\varnothing, \varnothing]$, and the total effect $\text{TE}(\tilde{t}_a) = \boldsymbol{Y}_{>\tilde{t}_a}[\tilde{t}_a, \tilde{t}_a] - \boldsymbol{Y}_{\tilde{t}_a}[\varnothing, \varnothing]$. We repeat the experiments 10 times, and report the mean and $\pm 1$ standard deviation in Figs. 13 to 15.

From Fig. 13, we see that the results are inconsistent when the same hidden confounding is applied to the training and test data. Even though the model makes biased predictions of meal dosages and responses, it is not penalized at test time since the test data has the same bias on average. In Figs. 14 and 15, the hidden confounding between the train and test data is different, i.e., the models are forced to make biased estimates.

In Figs. 14a and 15a, the MSE of $\text{NDE}(\tilde{t}_a)$ rises noticeably with the increased confounding level $\beta$. This is not surprising as the confounding biases both the post-surgery glucose response to meals (which affects $\boldsymbol{Y}_{>\tilde{t}_a}[\tilde{t}_a, \tilde{t}_a]$) and the post-surgery meal dosages (which affects $\boldsymbol{Y}_{>\tilde{t}_a}[\tilde{t}_a, \tilde{t}_a]$ and $\boldsymbol{Y}_{>\tilde{t}_a}[\varnothing, \tilde{t}_a]$). For our model, the MSE rises more pronouncedly compared to the baselines, because the latter were already misspecified to start with. In Figs. 14b and 15b, the MSE of $\text{NIE}(\tilde{t}_a)$ changes less with increased confounding compared to $\text{NDE}(\tilde{t}_a)$, as the post-surgery meal *times* are not affected by the confounding, and the dosage bias (which affects only $\boldsymbol{Y}_{>\tilde{t}_a}[\varnothing, \tilde{t}_a]$) affects the performance in $\text{NIE}(\tilde{t}_a)$ less compared to the response bias in $\text{NDE}(\tilde{t}_a)$. In Figs. 14c and 15c, the MSE in $\text{TE}(\tilde{t}_a)$ increases with confounding. However, this is less pronounced than for $\text{NDE}(\tilde{t}_a)$, as the confounding bias only affects $\boldsymbol{Y}_{>\tilde{t}_a}[\tilde{t}_a, \tilde{t}_a]$.

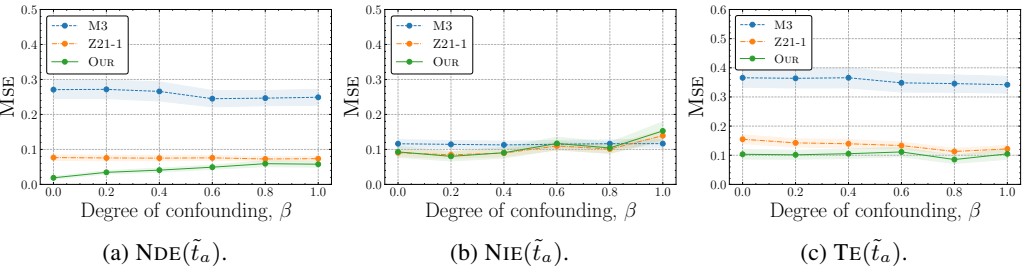

        (a) $\text{NDE}(\tilde{t}_a)$.           (b) $\text{NIE}(\tilde{t}_a)$.           (c) $\text{TE}(\tilde{t}_a)$.

Figure 13: MSE for NDE, NIE, TE for the confounding scenario (i), where the patients are advised to be active also in the *post-surgery* period in the test data, i.e., the same hidden confounding in training and test data.

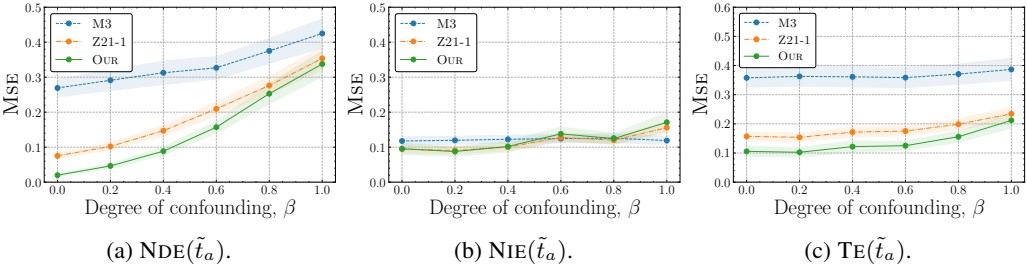

        (a) $\text{NDE}(\tilde{t}_a)$.           (b) $\text{NIE}(\tilde{t}_a)$.           (c) $\text{TE}(\tilde{t}_a)$.

Figure 14: MSE for NDE, NIE, TE for the confounding scenario (ii), where the patients are advised to be active in the *pre-surgery* period in the test data, i.e., different hidden confounding in training and test data.

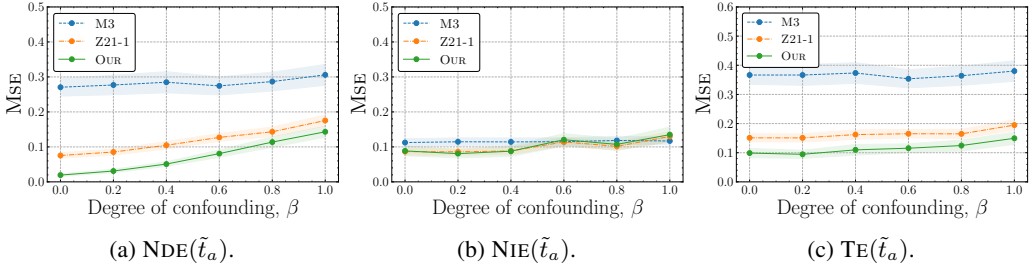

        (a) $\text{NDE}(\tilde{t}_a)$.           (b) $\text{NIE}(\tilde{t}_a)$.           (c) $\text{TE}(\tilde{t}_a)$.

Figure 15: MSE for NDE, NIE, TE for the confounding scenario (iii), where the patients are not advised to be active in the test data, i.e., hidden confounding in the training data, no hidden confounding in the test data.

