# OpenReview forum: "Temporal Causal Mediation through a Point Process: Direct and Indirect Effects of Healthcare Interventions"
_NeurIPS.cc/2023/Conference — NeurIPS 2023 poster_

### Official Review · Reviewer_5ke6 · 2023-07-06

**Soundness:** 3 good
**Presentation:** 3 good
**Contribution:** 3 good
**Rating:** 7
**Confidence:** 4

**Summary:**

This paper studies how to estimate the direct and indirect effctes of healthcare interventions. The general idea of this paper is to model the mediation process and outcome process jointly. More specifically, it considers the mediation process as a temporal point process conditioned on the past mediation, outcome and treatment data. It allows two causal paths: direct path models the direct effect of treatment and indirect path models the path treatment->mediation->outcome. The authors prove that under their three assumptions, the two effects can be represented by two terms in their non-paraametric temporal point process model. Experimental results shows the advantage of their proposed model.

**Strengths:**

(1) The proposed model studies an interesting problem: how to distinguish indirect and direct effect, which is important in healthcare

(2) The overall design of their causal model are reasonable. Althought their assumptions are not easy to verify on the data, they have tried their best to give convincing analysis to the data.

(3) The motivation is clear and convincing.

**Weaknesses:**

(1) Did you consider that in real world, the treatment may be correlated with the outcome, leading to bias in the model? Did you try to reduce the issue with IPTW or other method to debias?

(2) Some recent related works about handling treatment effect and causal inference with non-parametric temporal point process model are missing. For example, [1] studies the treatment effect in healcare, too. [2] considers how to debias the neural temporal point process in the context of social media analysis. [3] studies how to sample the counterfactual sequences from temporal point process.

[1] Gao, Tian, et al. Causal Inference for Event Pairs in Multivariate Point Processes. NeurIPS 2021

[2] Zhang, Yizhou, et al. Counterfactual Neural Temporal Point Process for Estimating Causal Influence of Misinformation on Social Media. NeurIPS 2022.

[3] Noorbakhsh, Kimia , and M. G. Rodriguez . Counterfactual Temporal Point Processes. NeurIPS 2022.

**Questions:**

Did you consider that in real world, the treatment may be correlated with the outcome, leading to bias in the model?

Did you try to reduce the issue with IPTW or other method to debias?

**Limitations:**

They have discussed.

---

> ### Author Rebuttal · Authors · 2023-08-09
>
> We are glad that you find our problem setting and approach both interesting and relevant. Thank you for highlighting the clarity of our motivation and the design of the causal model and analysis!
>
> **Regarding treatment–outcome correlation and potential bias:**
> In our real-world case-study, as discussed in the Causal Assumptions paragraph in Section 5.1, all patients undergo the surgery. Thus the probability of the treatment is independent of past outcomes. Hence, in this setup, there is no need to debias the effect estimations.
>
> However, if there were unobserved confounders affecting for example $Y$ and $A$, debiasing methods such as IPTW may of course be relevant. Thank you for making this important point, which we have added to the manuscript.
>
> **Regarding references on causal inference with temporal point processes:**
> Thank you for pointing out these references. We have now included them in the introduction.

---

> > ### Comment · Reviewer_5ke6 · 2023-08-12
> >
> > Ok, then I will keep my positive attitude.

---

### Official Review · Reviewer_NnCo · 2023-07-07

**Soundness:** 3 good
**Presentation:** 4 excellent
**Contribution:** 3 good
**Rating:** 5
**Confidence:** 4

**Summary:**

The paper aims to estimate the direct and indirect treatment effects of healthcare interventions. The authors model the mediator as a point process and propose a non-parametric mediator–outcome model where the mediator is assumed to be a temporal point process that interacts with the outcome process. The authors conduct experiments on a real-world RCT dataset and a semi-synthetic dataset. The experiments on the synthetic dataset show that the proposed model outperforms the baselines on the treatment effect estimation tasks.

**Strengths:**

- The authors propose a new approach to model the direct and indirect effects of healthcare interventions.
- The authors conduct experiments on both real-world and semi-synthetic datasets. The results demonstrate that the proposed model outperforms the baselines.
- The implementation code is available.


**Weaknesses:**

- Treatment A, as the confounder, affects both mediator M and outcome Y. M affects Y.
The confounding bias would make the estimation E[Y|A,M] inaccurate. Figure 5 shows the difference of diet density pre- and post-surgery, which demonstrates the existence of confounding bias. Without consideration of the confounders, the direct and indirect treatment effect estimation could be inaccurate.
- The authors claim that (A1, A2, A3) might not hold in observational studies and they are not statistically testable. I have the concern that if the assumptions do not hold, can the proposed model be applied to real-world applications? How to evaluate the potential risk of the model.
- A1，A2，A3 are very strong assumptions. In real-world settings, the no-unobserved confounder assumption may not hold, so it is necessary to conduct a sensitivity analysis of how sensitive or robust the proposed models are to the unobserved confounders.
- Bariatric surgery could cause weight loss. Another mediator weight would significantly change after the surgery.
- Some details are missing.  The authors just use diet and surgery to predict blood glucose. Is any detailed information about the diet, like nutrients including sugar, and starch? Are patients’ demographics (weights, age, height) used in the experiments?
- It would be better if the authors display the factual prediction performance, like MSE for glucose prediction.
- It is unclear what the variables m and o in Eq. (9) mean.


**Questions:**

See above.

---

> ### Author Rebuttal · Authors · 2023-08-09
>
> Thank you for your feedback and questions, which we hope to address in the following and with which we have improved the manuscript:
>
> **1. Confounding bias of treatment $A$ on mediator $M$ and outcome $Y$:**
> Please note that the estimation of $E[Y|A,M]$ is not confounded by $A$, because $A$ is included in the conditioning set. However, if there were unobserved confounders affecting for example $Y$ and $A$, then these confounders would bias the estimation. Our assumption A1 essentially is that there are no such unobserved confounders.
>
>
>
>
> **2. Application to real-world problems when causal assumptions A1, A2, A3 do not hold:**
> We agree that causal assumptions (A1, A2, A3) may not hold in real-world applications, that is why we included a discussion of causal assumptions in the real-world experiment in Section 5.1 and highlighted this as one of the main limitations. Note that this limitation applies to causality literature in general. Nevertheless, in our real-world study everybody undergoes the surgery, which rules out the possibility of a confounder affecting both the surgery and the outcome (A1).
>
>
>
>
>
> **3. Sensitivity analysis of NUC assumption:**
> The violation of the NUC assumption would result in an error in the causal effect estimates due to the unblocked back-door paths, whose magnitude depends on the size of the violation. We agree that an interesting future direction is to extend existing sensitivity analysis methods (Robins et al., 2000b) in order to estimate the size of the error in the present setting, but this is beyond the scope of the present article.
> Nevertheless, we decided to add another experiment to demonstrate the effect of hidden confounding, and combined this with *Reviewer bzqS*’s question about model misspecification. In the experiment, we created a hidden-confounding scenario where after the surgery, patients are advised to be more active. We sample an unobserved binary confounder for being active for each meal period with probability $\beta$. If active, a patient both (i) eats more, and (ii) their glucose peaks less. The results are shown in Figure 1 in the attached PDF, and shows a decline in estimation accuracy when the amount of hidden confounding increases. We are happy to take feedback during the author-reviewer discussion and improve this further towards the final version.
>
>
>
>
>
>
> **4. Handling of other mediators, e.g., weight loss:**
> As described in the *Dataset* paragraph of the experimental details in Appendix C.1, the post-surgery data is collected right after the surgery (within 2 weeks), and in this early period after the surgery, weight loss is not considered as a mediator [Bojsen-Møller+14]. Yet, this comment applies to all other possible mediators, and we agree that further discussion is beneficial.
>
> There are many possible mediators that could be added into the model, e.g., gut hormones, weight loss, hepatic insulin sensitivity, etc. In our real-world application, we focus on the dynamic causal mediation problem, where the sequence of meals is the mediator of the effect of surgery on blood glucose. In this definition, the rest of the metabolic processes, weight loss, or any other mediator through which the surgery could affect blood glucose, are included in the “direct effect”. An extension to more mediators and an explicit quantification of the corresponding indirect effects will be an interesting future direction, especially from a medical point of view, but beyond the scope of the current work.
>
>
> **5. Missing details on nutrient information and patient characteristics:**
> Thank you for pointing this out. The nutrient information was processed into carbohydrate amounts for each meal, equaling the sum of sugar and starch. Patients were non-diabetic individuals who had had bariatric surgery. More detailed patient characteristics were not available for our study due to data privacy. We have updated the manuscript accordingly and are happy to provide details during the discussion period if needed.
>
>
>
> **6. Factual prediction performance in MSE:**
> Could you please clarify what you mean by “factual prediction performance” and what you hope to learn from the results? Do you mean the MSE of the blood glucose over the entire test set trajectory, as predicted by each model? We have run this and show the results in Table 1 in the attached PDF. Note that this depends only on the outcome models and will not assess performance of the mediator models.
>
>
> **7. Clarification of m and o in Eq. 9:**
> They refer to the list of all past mediators and past outcomes, respectively. In our implementation, the mediator effect $g_m(\tau; \mathbf{m})$ depends on the relative times of the last $Q_m$ mediators (meals), and the outcome effect $g_o(\tau; \mathbf{o})$ depends on the relative times and values of the last $Q_o$ glucose outcome measurements, as described in Appendix B.3.1. We have clarified this in the manuscript.

---

> > ### Comment · Area_Chair_x3ao · 2023-08-18
> > **Response to rebuttal**
> >
> > Dear reviewer,
> >
> > The author rebuttal appears to have presented several targeted responses to your questions.
> >
> > Are your questions appropriately addressed?
> > If they are, would you consider re-assessing your score in light of them.
> > If not, please do provide additional context and feedback to the author.
> >
> > In either case, please provide an acknowledgement of the effort the authors put in, why your questions have (or have not) been addressed and what your assessment of the work is in light of this evidence with a view to reach consensus with the other reviewers on this work.
> >
> > -AC

---

> > ### Comment · Reviewer_NnCo · 2023-08-18
> >
> > Thanks for the comprehensive replies from the authors. I still have some concerns:
> >
> > Regarding the sensitivity analysis, it would greatly enhance the robustness of your study if you could also assess the model's performance when confounding influences impact both the training and test datasets. Given the realities of real-world scenarios, the presence of hidden confounders can significantly impact the test data, thereby warranting an exploration of this aspect.
> >
> > In relation to the factual prediction experiments, while I appreciate the utilization of a 2-day training data setup with the remaining day for validation, I would suggest a more patient-centric division of the training and test sets. By avoiding potential overfitting concerns due to shared patient data in both sets, your results would be further strengthened. Additionally, there is a need for clarification on whether cross-validation was conducted in the experiments presented in the original manuscript. If the experiment setup aligns with the factual prediction experiments outlined in the attached pdf, the persuasiveness of the outcomes may be compromised.
> >
> > It's worth highlighting that metabolic rates exhibit considerable variation among individuals, a phenomenon intricately tied to demographic factors such as age, gender, body weight, and height. These demographics could be potential confounding variables. Their absence could potentially introduce inaccuracies in the interpretation of treatment effects. Moreover, such demographic insights are integral to randomized controlled trials (RCTs), offering a lens to assess the representativeness of RCT data.

---

> > > ### Author Response · Authors · 2023-08-21
> > >
> > > Thanks for the insightful comments.
> > >
> > > **More comprehensive sensitivity analysis study:**
> > > For the final version, we are happy to extend our experiment with confounding in both training and test sets.
> > >
> > > **A more patient-centric division of the training and test sets for the factual prediction experiments:**
> > > Our model is hierarchical: the baseline and response magnitude are individual-specific, and the response shape is shared. We have not tried predicting patients with models trained purely on other patients, but the accuracy would likely be low due to relatively large differences between individuals. Predictions tuned for an individual are well-motivated also from the application point-of-view, and the added experiment reflects this.
> > >
> > > **Clarification on cross-validation in the experiments:**
> > > The most important hyperparameters were the GP lengthscales, which were selected by a combination of domain knowledge and validation of the model fit. For meal-response, we considered lengthscales 0.15 h, 0.3 h, 0.5 h, 1.0 h, and for the baseline, lengthscales 5 h, 10 h, 20 h. We’ll clarify the treatment of these and other hyperparameters in the Appendix.
> > >
> > > **If the experiment setup aligns with the factual prediction experiments outlined in the attached pdf, the persuasiveness of the outcomes may be compromised:**
> > > After contemplating this since the first rebuttal, we felt like we failed to emphasize that the main contribution is really on estimating direct and indirect effects, suitable for answering counterfactual questions: for example, what would happen if only the diet changed but not metabolic processes, or vice versa. For such questions, the factual outcomes are not available and therefore we feel that an experiment comparing the MSE between predictions and factual outcomes is inevitably a bit misaligned with the rest of the paper. On the other hand, we feel a more justified validation is obtained by measuring the accuracy of direct and indirect effect estimates in the semi-synthetic experiment, and benchmarking the estimates with domain knowledge in the real-world study. We apologize for not communicating this clearly in the earlier response but hope this clarifies now. Nevertheless, we will also include the factual prediction experiment in the Appendix as promised.
> > >
> > > **Potential confounders:**
> > > We agree and will further emphasize this as a limitation of the present study.

---

### Official Review · Reviewer_jr2N · 2023-07-08

**Soundness:** 3 good
**Presentation:** 2 fair
**Contribution:** 3 good
**Rating:** 5
**Confidence:** 3

**Summary:**

The paper defines direct and indirect effects in complex healthcare time-series as dynamic stochastic processes and theoretically provides causal assumptions for identifiability. This model allows for an external intervention influencing both mediator and outcome sequences simultaneously and captures time-delayed interactions among them.

**Strengths:**

1.	The authors proficiently present the estimated direct and indirect effects as longitudinal counterfactual trajectories, along with the requisite theoretical causal assumptions for their identification.
2.	The method proposed is neat and articulated with good clarity.

**Weaknesses:**

1.	The method's scalability to larger datasets poses a concern. There exists many methods specifically designed for high-dimensional mediation analysis in time series [1][2][3]. It would be enlightening to observe how the proposed method compares to these in handling complex datasets.
2.	To my understanding, Figure 1a may not accurately depict Zeng et al. 2021 [4]. It seems the original work allows for past mediators to have an influence on future outcomes.

References

[1] Chén, Oliver Y., et al. "High-dimensional multivariate mediation with application to neuroimaging data." Biostatistics 19.2 (2018): 121-136.

[2] Zhang, Haixiang, et al. "Mediation analysis for survival data with high-dimensional mediators." Bioinformatics 37.21 (2021): 3815-3821.

[3] Luo, Chengwen, et al. "High-dimensional mediation analysis in survival models." PLoS computational biology 16.4 (2020): e1007768.

[4] Zeng, Shuxi, et al. "Causal mediation analysis for sparse and irregular longitudinal data." The Annals of Applied Statistics15.2 (2021): 747-767.

**Questions:**

1.	What advantage does the utilization of a marked point process offer in modeling mediators compared to the approach in [1], where the observed mediator is considered as drawn from a smooth underlying process?
2.	Does the proposed method allow for modeling a high-dimensional observed mediator?
3.	In line 119-120, the mediator process M considers the number of occurrences of the mediating event up until time $\tau$ and the value of the mediator at time $\tau$. I'm curious if the occurrence count alone is sufficient to model the process, or if the past values of the mediator should also be considered?
4.	Regarding line 146, how is the continuation of the outcome after the intervention at $t_a$ represented? Should it be an equal average over all timepoints post $t_a$, or should it be a weighted average giving more importance to timepoints closer to $t_a$?
5.	For quick clarification, in line 159, $H_{\leq \tau}$ refers to the history up until time $\tau$. Does this history include both mediator and outcome?

Reference

[1] Zeng, Shuxi, et al. "Causal mediation analysis for sparse and irregular longitudinal data." The Annals of Applied Statistics15.2 (2021): 747-767.

**Limitations:**

Yes, the authors addressed their limitation, which involve untestable causal assumptions and scalability of the method to larger data sets.

---

> ### Author Rebuttal · Authors · 2023-08-09
>
> Thank you for appreciating the articulation of our proposed method and for your valuable questions. We hope we can further clarify and address your concerns in the following:
>
> **Regarding scalability:**
> There appears to be a misunderstanding, as the works referred to [Chen+18, Luo+20, Zhang+21] are not for time-series, but for the static causal mediation with a single, high-dimensional mediator. Hence, it is not possible to directly compare them to our method.
>
> We discuss scalability to a larger number of data points as a main limitation in the conclusion and provide further details in Appendix B. For standard Gaussian processes (GP), as in our outcome model, the computational complexity is $\mathcal{O}(N^3)$ where $N$ is the number of training data points. For GPs with inducing point approximations, as in our mediator model, the complexity is $\mathcal{O}(NM^2)$, where $M$ is the number of inducing points. Improving these is an important future research direction but falls outside the scope of the present work.
>
> Regarding the scalability to higher-dimensional mediators, there is no fundamental restriction on the dimensionality of the mediator dimensionality in our joint distribution in Eq. (7). As our main contribution lies in the formulation of the dynamic causal mediation problem, which is not restricted to just GP-based modeling, we could replace the GP-based mediator model for example with a neural ODE to handle high-dimensional dynamics. This also is beyond the scope of the current work and can be an exciting future direction.
>
> **Regarding accuracy of depiction of [Zeng+2021]:**
> We assume this refers to our Figure *2*a. To clarify, [Zeng+2021] “assume the effect of the mediation process on the outcome is concurrent, namely the outcome process at time $t$ does not depend on the past value of the mediation process” (their paragraph leading to Eq. (4.9); see also Eq. (4.9)). This is what we show in the figure. Nonetheless, [Zeng+2021] indeed include a potential influence from past mediators to future outcomes in their _theoretical_ framework, but this cannot be solved analytically. We have updated the caption of our figure to clarify this point.
>
>
> Regarding your remaining questions:
>
> **Q1: Comparison of point process mediators to high-dimensional mediators in [Zeng+2021], “where the observed mediator is considered as drawn from a smooth underlying process”:**
> The data we are considering are fundamentally discrete events, for which point processes are an appropriate model, and a smooth process would not be because it neglects the information in the event times. In other applications, where event times are non-informative, a smooth mediator process might be appropriate.
>
> **Q2: Allowing for a high-dimensional observed mediator?**
> Our method/problem formulation is agnostic to the choice of models. With the current setup including a GP-based mediator model, modeling high-dimensional mediators such as images may not be feasible. However, this could be an interesting future direction and for example, a suitable model such as a neural ODE could be used as the mediator mark intensity to capture high-dimensional dynamics.
>
> **Q3: Clarification of the Mediator Process definition (line 119-120):**
> We condition on the history of the whole process, which actually also includes the previous mediator values. Hence, our model does consider the past mediator values as well. We thank you for pointing out this aspect that would benefit from additional clarification, and we have improved the text correspondingly.
>
> **Q4: Clarification of the Outcome Process definition (line 146):**
> The outcome process definition $Y_{>t_a} = \{Y(\tau): \tau > t_a\}$ is the stochastic process defined on the interval $[t_a, T]$ as described by the fitted model. As such it describes the *entire* future trajectory, not a single value (such as an average over time points). We have made this more explicit in the manuscript.
>
> **Q5: History definition (line 159):**
> Yes, it includes both mediators and outcomes. We have now made this explicit in the manuscript.

---

> > ### Comment · Area_Chair_x3ao · 2023-08-18
> > **Response to rebuttal**
> >
> > Dear reviewer,
> >
> > The author rebuttal appears to have presented several targeted responses to your questions.
> >
> > Are your questions appropriately addressed?
> > If they are, would you consider re-assessing your score in light of them.
> > If not, please do provide additional context and feedback to the author.
> >
> > In either case, please provide an acknowledgement of the effort the authors put in, why your questions have (or have not) been addressed and what your assessment of the work is in light of this evidence with a view to reach consensus with the other reviewers on this work.
> >
> > -AC

---

> > ### Comment · Reviewer_jr2N · 2023-08-18
> >
> > Thank you for the comprehensive clarification on various points that I raised in my review. I appreciate the time and effort you put into addressing my concerns, and  I find it justifiable to marginally increase my rating to a 5 and my confidence score to 3.
> >
> > I have one follow-up question on the choice of models. Since the method and problem formulation are agnostic to the choice of models, have you considered or conducted any empirical studies to validate the model's robustness across different model choices?
> >
> > Overall, I look forward to seeing the final version of this manuscript, as temporal causal mediation is a very intereting field with great application potential. However, I am really interested in how this might scale, especially since high-dimensional observed mediators are getting more and more prevalent in many applications. Exploring this could uncover even more impactful insights and opportunities.

---

> > > ### Author Response · Authors · 2023-08-21
> > >
> > > Thanks for your thoughtful comment.
> > >
> > > **Different modeling choices:**
> > > With the synthetic datasets, we considered multiple choices for the mediator and outcome models (Table 2). In the real world case-study, we used our complete model, and in the new real-world results added in the rebuttal we also included alternatives, though neural ODE based approaches were not included in this study.

---

### Official Review · Reviewer_bzqS · 2023-07-26

**Soundness:** 4 excellent
**Presentation:** 4 excellent
**Contribution:** 2 fair
**Rating:** 7
**Confidence:** 3

**Summary:**

The paper's outstanding qualities lie in its well-articulated presentation and its precise experimental design. It amalgamates the earlier research findings of [Zeng et al., 2021] and [Hızlı et al., 2022] with the innovative notions put forth by [Robins et al., 2022] on indirect effects. The authors tackle pragmatic issues, such as the effects of surgery on a patient's blood sugar levels in relation to their diet, by formulating pertinent questions. For instance, they question whether optimal post-surgery mediation can entirely regulate blood sugar levels, or if there exist uncontrollable surgery-induced effects on blood sugar levels that resist management through mediation and diet adjustments.

In their methodological approach, the authors utilize non-parametric models of the temporal data-generating process. They employ a Marked Point Process (MPP) akin to [Hızlı et al., 2022] for meal intake (the mediating factor) and a non-parametric Gaussian Process (GP) for the outcome. The former is modeled as a combination of a counting process (number of meals) and a dosage process (carb intake per meal) using a non-parametric Poisson Process and the latter as a Gaussian process. The mediator model is trained to predict the mediator based on the intervention and the outcome model is trained with both the mediator and intervention given as input. This approach is put to the test in a series of synthetic experiments, validating its predictive strength against baseline models like Zeng et al., 2021. They further apply it in a real-world setting, successfully reproducing biological insights and potentially addressing the question of the degree to which surgery impacts diet changes.

While at first glance, this paper might seem to bear similarities to [Hızlı et al., 2022], it stands out through its adept combination of the best methods for examining direct and indirect cause-and-effect relationships in a temporal setting. The paper, with its organized code and clear writing style, has the potential to become a valuable asset. However, I'm leaning towards accepting this paper on the condition that certain concerns regarding its originality and novelty are addressed, which I will detail in the following.

**Strengths:**

1. The paper is commendable for its realistic problem setup, which is articulated in Section 5.1 dealing with corner cases where the assumptions might falter: the existence of hidden confoundings and the violation of assumptions A.1 to A.3. Such validation is crucial in causal studies to confirm assumptions and identify any unnoticed confoundings that may influence our conclusions. Furthermore, the experiments and predictions are consistent with the clinically significant direct and indirect impacts of bariatric surgery on blood glucose levels.
2. The approach to modeling the temporal dynamic is robust, anchoring its foundation on recent, proven work that adds to its credibility.
3. The paper's eloquent presentation is worthy of note. The reading experience is enhanced by effective use of color-coding to differentiate between mediator and direct interventions. A minor suggestion would be to consider adaptations for grayscale printed versions of the paper. For instance, the caption of Figure (2) includes light and dark blue color coding, which could be made more distinguishable by slightly altering the arrow patterns.

**Weaknesses:**

1. The theoretical advancement of the study appears relatively marginal. While the exploration of direct vs. indirect causal effects in a temporal setting is engaging and the experiments provide valuable insights, I have some reservations about two of the claimed main contributions:
* Dynamic causal mediation with a point process mediator: [Hızlı et al., 2022] have previously introduced point process mediator modeling. The novelty here is questionable, given that in the prior work, the treatment was the mediator itself.
* A mediator-outcome model with an external intervention: The distinctiveness here is the training of two models: pre-intervention and post-intervention. However, the applied intervention is overly simplistic, offering limited theoretical innovation or insight. I have proposed, in the "Questions" section, the inclusion of the theory behind more complex interventions and experimentation on the simpler case. Yet, as it stands, this contribution mainly replicates the approach from [Hızlı et al., 2022], but uses two models to account for the intervention.
2. Table 1 presents results suggesting that the direct causal impact of surgery outweighs its indirect effects. Although the insights from 5.1.3 and 5.1.2 align with existing studies, there seems to be no supportive evidence for this hypothesis. Perhaps incorporating relevant literature explanations into the discussion would be beneficial. While the coherence between findings in 5.1.2 and 5.1.3 lend some validation to the model, it would still be advantageous to have literature support for 5.1.4.

**Questions:**

1. Even though the paper presents a succinct and coherent narrative, it's hard to ignore that the methodology could easily extend to cases where the intervention itself is also a point process. For instance, one might consider a patient's long-term history and periodic clinical treatments. In such scenarios, $NIE$ and $NDE$ could be defined at different time points. It might be beneficial to include the theory behind this in the appendix section. The theoretical framework in sections 2 and 3 would work if one defines $N_A: [0, T] \to \mathbb{N}$, and instead of developing two distinct models, a more comprehensive model could be formulated that includes the history of interventions. This approach could also minimize the chance of future incremental papers being published.
2. What is the model's predictive power under model misspecification? Currently, the paper only presents results assessing predictive power in semi-synthetic scenarios where the model is appropriately specified. However, it would be beneficial to conduct experiments in scenarios reflective of real-world settings where model misspecification is common. Although the paper does incorporate a real-world setting, it is used merely for extracting qualitative observations. A comparative analysis of prediction power, similar to the semi-synthetic scenario but including model misspecification, is missing.

**Limitations:**

1. Like any causality-oriented study, this paper relies on certain assumptions for its model to function effectively. However, the authors have acknowledged this constraint adequately. They have also indicated possible real-world scenarios with hidden confounders where these assumptions might be compromised. This demonstrates a good understanding of the model's limitations.
2. A further limitation of the method, as previously hinted, is its inability to handle more complex interventions. Currently, the model considers total effect, direct, and indirect effect only in relation to a unitary, binary intervention point process. It could potentially be enhanced by extending its capacity to accommodate non-binary scenarios (for different intervention styles, such as dosage treatments) or non-unitary processes (considering the entire electronic health record of a patient over a long period), which could provide a more comprehensive understanding.

---

> ### Author Rebuttal · Authors · 2023-08-09
>
> Thank you for your thorough feedback! In the following we hope to address your concerns regarding originality and novelty:
>
> **Dynamic causal mediation with point process mediator:**
> We emphasize that the whole formulation of the problem with point process mediator and outcome, as well as defining what direct and indirect effects mean in this context and proving their identifiability, are novel contributions, even if the model used in the estimation builds on existing works. The significant difference between our work and [Hizli+2022] is the problem that we are solving, namely dynamic causal mediation. In [Hizli+2022], while the treatments could be considered as mediators, there are no direct arrows from the intervened variable (the policy) to the outcomes. The direct/indirect effect estimation problem with the included direct arrows is fundamentally harder and even unidentifiable within the potential outcome based formulation of [Hizli+2022], when the mediator–outcome processes interact. Notice that [Zeng+2021] exclude arrows from the past outcomes to the future mediators due to the same reason. We solve this by following the interventionist approach to causal mediation [Robins+22].
>
> **Extension to more complex interventions:**
> Following your suggestion, we have extended the current problem setup to a sequence of (hard) interventions, possibly containing non-binary or non-unitary treatments, and will add this to the text. Here, we only briefly specify the differences compared to Section A.2 and provide the final result. We are happy to provide further details during the discussion period.
>
> We define the direct, indirect, and total causal effects as differences in the potential outcomes with respect to two intervention sequences: (i) a target treatment sequence $\mathbf{a}^1 = \{a^1_1, \ldots, a^1_{N_1}\}$ (similar to the surgery in the current setup), and (ii) a reference treatment sequence $\mathbf{a}^0 = \{a^0_1, \ldots, a^0_{N_0}\}$ (similar to the “no treatment” in the current setup). To answer the causal queries, we need to identify the counterfactual trajectory $\mathbf{Y}[\text{direct path}=\mathbf{a}^0, \text{indirect path}=\mathbf{a}^1]$, which is the outcome under the intervention that sets the direct-path treatment sequence to $\mathbf{a}^0$ and the indirect-path treatment sequence to $\mathbf{a}^1$. The final result is
>
> $$P(Y_{\mathbf{q}}[\mathbf{a}^0, \mathbf{a}^1]) = \sum_{M_{ > t_a}} \prod_{r=0}^{R-1} P(Y_{q_{r+1}} | \mathbf{A} = \mathbf{a}^0_{ < q_{r+1}}, M_{[q_r, q_{r+1})}, \mathcal{H}_{\leq q_r}) \quad $$
>
> $$\qquad\times P(M_{[q_r, t_p]} | \mathbf{A} = \mathbf{a}^1_{ < q_r}, \mathcal{H}_{\leq q_r}) $$
>
> $$\qquad\times P(M_{(t_p, q_{r+1})} | \mathbf{A} = \mathbf{a}^1_{\leq t_p}, M_{[q_r, t_p]}, \mathcal{H}_{\leq q_r})$$
>
> $t_p$ represents a possible treatment time in the sequence $\mathbf{a}^1$ that falls in the interval $[q_r, q_{r+1}]$. Different from Eq. (7) of our submission, to account for such a treatment $a_p=(t_p,d_p)$, we divide the mediator process into two parts, both conditioned on the treatments up to the start of their interval. Without loss of generality, we consider only a single treatment in the interval $[q_r, q_{r+1})$ at time $t_p$. It can easily be extended to multiple treatments, leading to further factorization of the mediators in the interval. As you also suggested, this implies that mediator and outcome models should depend on the history of the intervention sequence. This can be either (i) included in the current GP framework by concatenating a summary of the history to the kernel inputs, or (ii) modeled with a more expressive model such as a neural ODE.
>
> If the treatment sequences are further considered as a set of stochastic (soft) interventions in continuous time similar to [Hizli+22], then this can be considered as a point process. Accordingly, the above result can be further extended, similar to the extension of [Schulam+17] to [Hizli+22]. This can be achieved by averaging the above result over all possible treatment sequences, i.e., adding the probability of observing the treatment sequence in each interval to the above equation and integrating out the whole treatment sequence.
>
> **Literature support for 5.1.4:**
> Prompted by your comment, we delved deeper into the medical literature and found Laferrère et al., The Journal of Clinical Endocrinology & Metabolism 93 (7), 2008, who study the effects of two different interventions—namely, dietary changes and surgical procedures—on individuals who had lost equal amounts of weight (10 kg). The study focuses on the impact of a 50 g glucose dose, notably showcasing a more pronounced decline, particularly towards the conclusion of the test, in the surgery group compared to the control group. Because all patients received the same meal, this decline corresponds to the direct effect. This direct effect can be attributed to a range of intricate hormonal influences, including the involvement of gut hormones like GLP-1, alongside heightened insulin peaks subsequent to surgical intervention.
>
> Hence, application-wise, the novelty of our work lies in its exploration, by computational means, of the contribution of dietary change vs. predicted metabolic changes within a person. To our knowledge, there have been no existing models for this, making our work a pioneering contribution. We have added this reference and discussion in Section 5.1.4.
>
> **Predictive power under model misspecification:**
> The benchmarks in our semi-synthetic study already include misspecified models with respect to the ground-truth simulator (which is the same as the most complex model). Our empirical findings (Table 2) suggest that the performance deteriorates proportional to the amount of misspecification. Nevertheless, we agree that it is good to study model misspecification more systematically. We combined this idea with *Reviewer NnCo*’s question (Q3) about the performance under hidden confounding. Please see our response there.

---

> > ### Comment · Reviewer_bzqS · 2023-08-16
> >
> > I would like to thank the authors and acknowledge reading their response.

---

> > > ### Comment · Reviewer_bzqS · 2023-08-18
> > >
> > > Considering the underlying theory of the extension, along with the added support for version 5.1.4, I find it justifiable to marginally increase my rating to a 7. While I still perceive the theory and methodology as somewhat incremental, the compelling practical results that back them up, paired with the comprehensive responses provided, lead me to believe that an increase in score is warranted.

---

> > > > ### Author Response · Authors · 2023-08-22
> > > >
> > > > Thank you again for your considerate feedback and raising your score of our submission!

---

### Author Rebuttal · Authors · 2023-08-09

We thank all reviewers for their time and effort in writing thoughtful and valuable reviews. Overall, the reviews appear positive about our submission. We are happy for the explicit commendations of our realistic and relevant problem setup, robust modeling approach, and clarity of our presentation and experimental design.

In response to the reviews, we have extended our framework to the more general case where the intervention is itself a sequence of treatments (see response to Reviewer bzqS). Furthermore, we added a new experiment to study the effect of unobserved confounders and model misspecification (see response to Reviewer NnCo). We have clarified the manuscript as described in the individual responses thanks to the feedback and questions from all reviewers.

We hope we have adequately addressed all open questions in the individual responses and are looking forward to the discussion period.

---

### Decision · Program_Chairs · 2023-09-21

**Decision:**

Accept (poster)

**Comment:**

This paper uses models to capture and correct for the direct and indirect effects of time-varying interventions. The work models the mediation process as a temporal point process conditioned on the past mediation and a non-parametric Gaussian Process for the outcome model (conditional on the mediator). The mediator model is trained to predict the mediator conditional on intervention and the outcome model is trained using the mediator and intervention as input. The model is evaluated in synthetic data against several existing baselines using a semi-synthetic dataset where its efficacy is verified after which it is evaluated on real data.

The reviewers, on average found the work well written, easy to follow, and even if incremental in terms of theoretical advance, felt the work presented a useful practical example of leveraging ML tools for capturing treatment effects in the presence of mediators in time series data. During the author discussion phase, the reviewers request additional experiments assessing the degree to which the model was affected by unobserved confounding -- the authors ran experiments in the form of sensitivity analysis studying how their treatment effects varied due to weaker and stronger confounding.

These results should be incorporated and discussed in the final revision. In addition to the other fixes detailed in the discussion period, as discussed with Reviewer NnCo, please do also include the variant of the experiment and the effect of confounding at evaluation time (test data).